# Environmental controls of rapid terrestrial organic matter mobilization to the western Laptev Sea since the Last Deglaciation

**Tsai-Wen Lin[1,2], Tommaso Tesi[3], Jens Hefter[1], Hendrik Grotheer[1,2,4], Jutta Wollenburg[5], Florian Adolphi[1,2,4], Henning A. Bauch[6,7], Alessio Nogarotto[3], Juliane Müller[2,4,6], and Gesine Mollenhauer[1,2,4]**

[1]Marine Geochemistry Section, Alfred Wegener Institute, Helmholtz Centre for Polar and Marine Research, 27570, Bremerhaven, Germany

[2]Department of Geosciences, University Bremen, 28334, Bremen, Germany

[3]Institute of Polar Sciences, National Research Council, 40129, Bologna, Italy

[4]MARUM Centre for Marine Environmental Sciences, University Bremen, 28334, Bremen, Germany

[5]Marine Biogeosciences Section, Alfred Wegener Institute, Helmholtz Centre for Polar and Marine Research, 27570, Bremerhaven, Germany

[6]Marine Geology Section, Alfred Wegener Institute, Helmholtz Centre for Polar and Marine Research, 27570, Bremerhaven, Germany

[7]Research Division 1: Ocean Circulation and Climate Dynamics, GEOMAR, Helmholtz Centre for Ocean Research, 24148, Kiel, Germany

**Correspondence:** Tsai-Wen Lin (tsai-wen.lin@awi.de) and Gesine Mollenhauer (gesine.mollenhauer@awi.de)

**Abstract.** Arctic permafrost stores vast amounts of terrestrial organic matter (terrOM). Under warming climate conditions, Arctic permafrost thaws, releasing aged carbon and potentially impacting the modern carbon cycle. We investigated the characteristics of terrestrial biomarkers, including *n*-alkanes, fatty acids, and lignin phenols, in marine sediment cores to understand how the sources of terrOM transported to the ocean change in response to varying environmental conditions, such as sea-level rise, sea-ice coverage, inland climate warming, and freshwater input. We examined two sediment records from the western Laptev Sea (PS51/154 and PS51/159) covering the past 17.8 kyr. Our analyses reveal three periods with high mass accumulation rates (MARs) of terrestrial biomarkers, from 14.1 to 13.2, 11.6 to 10.9, and 10.9 to 9.5 kyr BP. These terrOM MAR peaks revealed distinct terrOM sources, likely in response to changes in shelf topography, rates of sea-level rise, and inland warming. By comparing periods of high terrOM MAR in the Laptev Sea with published records from other Arctic marginal seas, we suggest that enhanced coastal erosion driven by rapid sea-level rise during meltwater pulse 1A (mwp-1A) triggered elevated terrOM MAR across the Arctic. Additional terrOM MAR peaks varied regionally. Peaks from the Beaufort Sea during the Bølling–Allerød coincided with a freshwater flooding event, while peaks from the Laptev Sea and the Fram Strait during the Preboreal/early Holocene coincided with periods of enhanced inland warming and prolonged ice-free conditions. Our results highlight the influence of regional environmental conditions, in addition to global drivers, which can either promote or preclude regional terrOM fluxes.

## 1 Introduction

Circumarctic permafrost, the ground that has remained below 0 °C for at least 2 consecutive years, plays an important role in the Arctic carbon cycle and is strongly affected by

the rapid warming in the Arctic, which is greater than the average warming of the Northern Hemisphere (Harris et al., 1988; Miller et al., 2010; Rantanen et al., 2022). It is one of the main terrestrial carbon sinks at present (Hugelius et al., 2014; Strauss et al., 2017). However, global warming could lead to permafrost thawing, potentially shifting these areas from a carbon sink to a carbon source (Winterfeld et al., 2018; Lara et al., 2019; Laurent et al., 2023). Due to the low ground temperature, organic matter stored in permafrost soils remains protected from degradation and release into the modern carbon cycle (Harris et al., 1988; Hugelius et al., 2014; Strauss et al., 2017). When permafrost thaws, the organic matter is transported through fluvial networks and coastal erosion to the ocean, eventually being deposited in marine basins (Schuur et al., 2008; Zhang et al., 2022). In addition to the degradation at the site of thawing, during transport, previously freeze-locked organic matter can be decomposed by microbes, releasing greenhouse gases such as $CO_2$ and $CH_4$.

The extent of organic matter degradation from thawing permafrost varies between sources and transportation trajectories (Vonk and Gustafsson, 2013; Strauss et al., 2015). Permafrost deposits from different depths (and ages) are mobilized via different pathways. Initially, increasing land temperatures and precipitation mobilize surface permafrost by deepening the active layer and expanding wetland areas. In later stages, the development of thermokarst and taliks exposes and mobilizes deeper permafrost layers (Schuur et al., 2008). Accelerated coastal erosion facilitates the transport of deep permafrost from cliffs to the ocean, particularly in regions with high cliffs such as the Siberian coast (Vonk and Gustafsson, 2013). However, permafrost mobilization caused by accelerated coastal erosion can be mitigated by the presence of land-fast sea ice (Rachold et al., 2000; Overduin et al., 2016; Nielsen et al., 2020; Irrgang et al., 2022). Understanding the dominant pathways of permafrost mobilization under different environmental conditions is critical for evaluating the response of Arctic permafrost to future warming. Changes in terrestrial organic matter (terrOM) mobilization and environmental conditions can be recorded in marine sedimentary archives.

Palaeoclimate records from the Last Deglaciation to the early Holocene (between ca. 19 and 11 kyr BP) offer insights into Arctic permafrost changes in response to warming and rising sea levels (Clark et al., 2012). During the Last Deglaciation, atmospheric $CO_2$ concentrations increased, temperature increased globally and amplified in the Arctic, and global sea levels rose, including rapid events and meltwater pulses (Shakun et al., 2012; Lambeck et al., 2014; Marcott et al., 2014; Köhler et al., 2017). Rapid increases in atmospheric $CO_2$ concentration occurred at 16.4, 14.6, and 11.5 kyr BP. The 14.6 kyr BP event coincided with an accelerated global sea-level rise and a period of reduced Arctic sea-ice cover, while the 11.5 kyr BP event coincided with intensified inland warming and a drop in sea-ice cover in the

Arctic (Fahl and Stein, 2012; Lambeck et al., 2014; Müller and Stein, 2014; Brosius et al., 2021; Detlef et al., 2023). Studying these periods of rapid environmental change can improve the understanding of how current abrupt warming, sea-ice loss, and sea-level rise might affect permafrost stability and the release of previously freeze-locked carbon.

Marine sedimentary archives from the Laptev Sea covering the early period of the Last Deglaciation (> 14 ka) are scarce, and existing records often have low temporal resolution and are discontinuous (Tesi et al., 2016a; Keskitalo et al., 2017; Martens et al., 2019, 2020). Here, we present two high-resolution sediment core records that, collectively, have continuously covered the last 17.8 kyr. We characterize the properties of organic matter exported from Siberian permafrost and relate changes in terrestrial carbon sources to respective climatic conditions. Results were then compared with published studies dealing with the terrOM released into the Arctic Ocean. Furthermore, we explore the interplay of potential factors driving rapid terrestrial carbon translocation, including sea-level fluctuations, inland permafrost stability, and variations in sea-ice cover.

## 2 Study area

The Laptev Sea is a marginal sea of the Arctic Ocean located north of Siberia (Fig. 1). The average depth of the Laptev Sea shelf is around 50 m, with a sharp shelf break located between 70 and 100 m water depth. Several sediment-filled palaeo-river channels cut through the shelf (Kleiber and Niessen, 2000). Due to its shallow water depth and flat topography, the Laptev Sea shelf is highly sensitive to sea-level fluctuations. During the Last Glacial Maximum, the Laptev Sea shelf was not covered by an ice sheet and was exposed during the low sea-level stand (Hughes et al., 2016), allowing the accumulation of permafrost deposits. The majority of the shelf was inundated between 12 and 6.5 kyr BP, with the rate of inundation slowing down thereafter (Jakobsson et al., 2012; Klemann et al., 2015) (Fig. 2h). This rapid shelf inundation led to southward shifts in both the coastline and the locations of major sediment deposition, causing a significant reduction in sediment input to the outer shelf (Bauch et al., 1999; Mueller-Lupp et al., 2000; Bauch et al., 2001). Today, in the Laptev Sea, sediments from water depths < 10 m are transported further offshore, and the primary deposition centre is located at around 30 m water depth (Kuptsov and Lisitsin, 1996; Are et al., 2002). The Laptev Sea plays a crucial role in the Arctic climate through extensive heat exchange with the atmosphere in summer and sea-ice formation in winter, making it a key region for studying the boundary conditions of the Arctic environment's response to climate change (Rudenko et al., 2014; Liu et al., 2022).

The Laptev Sea shelf receives substantial sediment inputs from both coastal erosion and river discharge (Lantuit et al., 2011; McClelland et al., 2016). Due to high cliffs and inten-

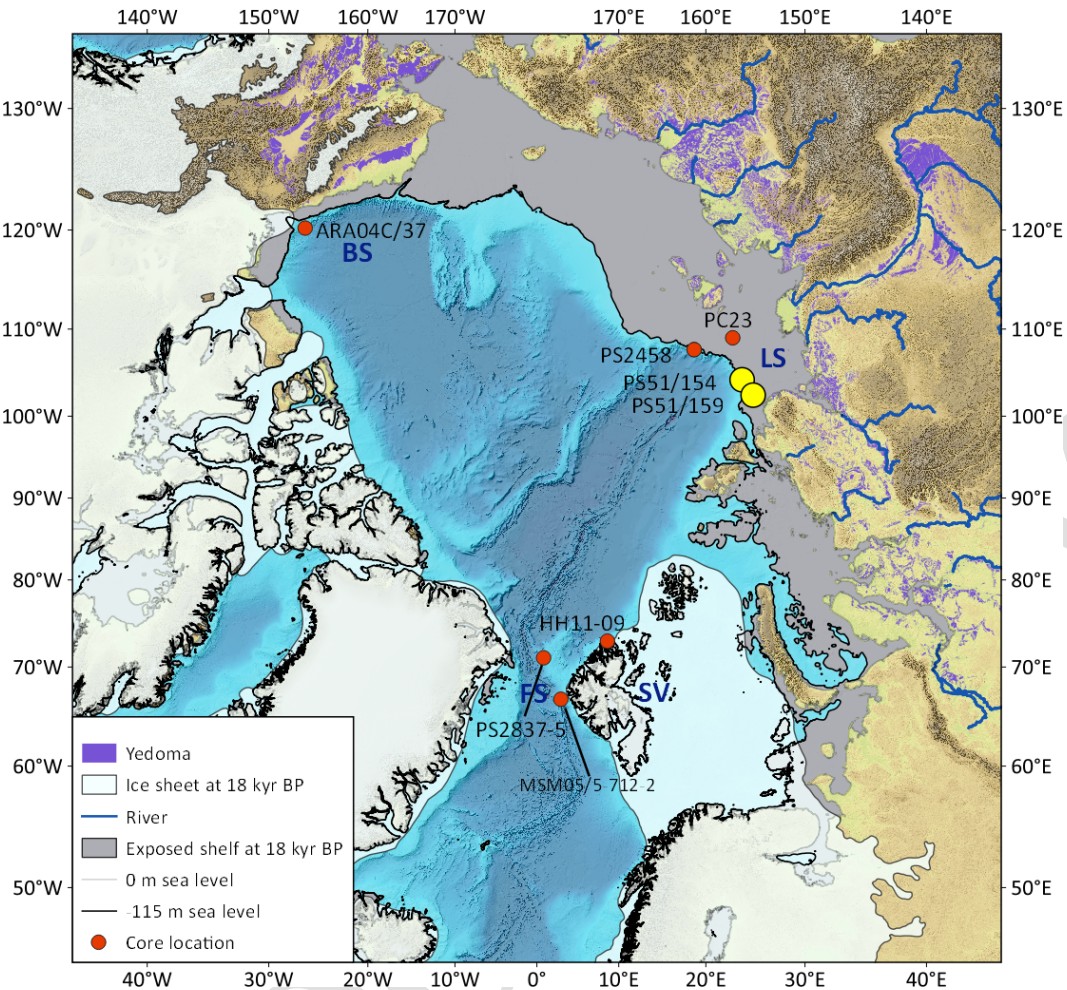

**Figure 1.** Map of the Arctic Ocean. The names of the marginal seas and archipelago are indicated by acronyms (BS: Beaufort Sea; LS: Laptev Sea; SV: Svalbard archipelago; FS: Fram Strait). The purple area indicates the modern Yedoma domain (Strauss et al., 2016, 2021, 2022). The white area shows the ice sheet cover at 18 ka (Dyke et al., 2003; Hughes et al., 2016). Deep-blue lines indicate the main river streams in the Arctic (Lehner and Grill, 2013). The exposed continental shelf area at 18 kyr BP is labelled in grey. The black line is the −115 m contour line, which is approximately the sea level in the early period of the Last Deglaciation (18 kyr BP) (Klemann et al., 2015). Yellow dots show the cores from the Arctic used in this study, including cores PS51/154 and PS51/159. Red dots show the other cores from previous studies that are used for comparison, including cores ARA04C/37 (Wu et al., 2020), PC23 (Tesi et al., 2016b), PS2458 (Spielhagen et al., 2005), HH11-09 (Nogarotto et al., 2023), PS2837-5 (Birgel and Hass, 2004), and MSM05/5-712-2 (Aagaard-Sørensen et al., 2014; Müller and Stein, 2014; Zamelczyk et al., 2014). The map background is from the International Bathymetric Chart of the Arctic Ocean (IBCAO) (Jakobsson et al., 2012). This map is plotted with QGIS3.22.9 (QGIS Geographic Information System; QGIS Association).

sified wave-induced erosion resulting from summer sea-ice melt, the Laptev Sea receives a large proportion of terrestrial material from coastal erosion. Erosion rates along the Laptev Sea coast are higher ($0.7\,\mathrm{m\,yr^{-1}}$) compared to the Arctic coastal average ($0.5\,\mathrm{m\,yr^{-1}}$) and may even reach more than $10\,\mathrm{m\,yr^{-1}}$ (Rachold et al., 2000; Lantuit et al., 2011; Günther et al., 2015). Additionally, the Lena River serves as the primary source of riverine sediments (McClelland et al., 2016; Holmes et al., 2021), but coastal erosion contributes more than twice the amount of terrestrial export compared to riverine sources (Rachold et al., 2000).

Most of the water discharged by the Lena River is transported eastward and northward, driven by local winds, carrying suspended sediments in the same direction, and guided by the sea bottom relief (Dmitrenko et al., 1999; Guay et al., 2001). In summer, drift ice and river discharges trigger high productivity of the Laptev Sea shelf (Ovsepyan et al., 2015; Hörner et al., 2016). Large amounts of terrigenous suspended particles in the Laptev Sea are incorporated into the sea ice (Dethleff, 2005). This terrigenous-material-rich sea ice is transported to the Fram Strait by the Transpolar Drift, accounting for 20 % of the total sea-ice flow to the Fram Strait (Rigor and Colony, 1997; Stein and Macdonald, 2004).

# 3   Materials and methods

## 3.1   Materials

Marine sediment cores PS51/154-11 (77.276° N, 120.610° E; water depth 270 m; abbreviated as PS51/154) and PS51/159-10 (76.768° N, 116.032° E; water depth 60 m; abbreviated as PS51/159) were obtained with Kasten corers during the ARK-XIV/1b (PS51 Transdrift-V) expedition in August 1998 aboard the R/V *Polarstern* (Kassens, 2016). Core PS51/154 was taken from the upper continental slope, and core PS51/159 was taken from the outer continental shelf (Fig. 1). After retrieval, the sediment cores were stored at $-20\,°C$ and subsampled at $4\,°C$. Subsamples were collected from the cores with 10 mL syringes in 5–10 cm intervals. The sampling intervals in core PS51/154 corresponded to an average temporal resolution of approximately 120 yr before 10 kyr BP, and, for core PS51/159, the average temporal resolution was about 270 yr. Due to the spatial proximity and consistency of biomarker proxy records between the two cores, we used the PS51/159 record to complement the low temporal resolution in core PS51/154 during the Holocene.

## 3.2   Microfossil radiocarbon dating and age–depth model

The age–depth models for cores PS51/154 and PS51/159 are based on microfossil $^{14}$C ages, initially established by Bauch et al. (2001), modified by Taldenkova et al. (2010) and Hörner et al. (2016), and refined in this study. Six new microfossil radiocarbon dates from core PS51/154 and one from core PS51/159 were added to the existing data sets (Table 1). Sediments were wet-sieved through a 63 μm mesh, and bivalve shell fragments were picked from the > 63 μm and oven-dried sediment fraction. Bivalve shells were examined under a microscope for species identification and photographic records (Table 1). Samples containing more than 250 μg C were converted to graphite targets, while samples containing less than that were analysed as $CO_2$ gas using the gas interface system (Table 1). Both types of analysis were conducted using the MIni CArbon Dating System (MICADAS) accelerator mass spectrometer at the Alfred Wegener Institute (AWI) (Mollenhauer et al., 2021). We used the Marine20 curve for calibration (Heaton et al., 2020), with a time constant $\Delta R = -95 \pm 61$ yr calculated using the Marine20 database (http://calib.org/marine/, last access: 9 August 2024). This $\Delta R$ value was derived from the average reservoir ages of five modern bivalve shells in the Laptev Sea (Bauch et al., 2001). The rationale for using an updated $\Delta R$ is the $\sim 150$-year shift in the global marine reservoir age between the Marine13 curve used in the previous age model (Hörner et al., 2016) and the Marine20 curve used in this study (Heaton et al., 2020; Heaton et al., 2023). The dating results are reported as calibrated ages (cal. yr BP). Age–depth models were constructed using the software OxCal 4.4 (Bronk Ramsey, 2009), with low-model-agreement data points identified as outliers and removed from the age–depth model. The outlier data are still shown in Table 1 and Fig. S1 in the Supplement TS1.

## 3.3   Bulk analysis

Stable carbon isotope compositions of total organic carbon ($\delta^{13}$C) in cores PS51/154 and PS51/159 were analysed using 15 mg of freeze-dried, homogenized sediment acidified with 1.5 M HCl in silver boats to remove carbonate. The acidified samples were then dried in the oven at 55 °C. $\delta^{13}$C values were measured using a Thermo Scientific FLASH 2000 CHNS Analyser coupled with a Thermo DeltaQ IRMS via CONFLO IV at the Institute of Polar Sciences, National Research Council (CNR-ISP). $\delta^{13}$C values were reported in parts per thousand (per mil, ‰) relative to Vienna Pee Dee Belemnite (VPDB). The standard error for replicate analyses of in-house standards was less than 0.15‰.

## 3.4   Lipid extraction and analysis

The lipid extractions were done following the procedure by Winterfeld et al. (2018). Freeze-dried, homogenized sediment samples were weighed at $1-5$ g and subsequently submerged in 25 mL dichloromethane (DCM) + methanol ($v/v = 9:1$) followed by ultrasonication (15 min) three times to acquire the total lipid extract (TLE). An internal quantification standard containing 889.6 ng squalane and 1558 ng 19-methylarachidic acid was added before extraction. After drying the TLE under a stream of nitrogen, the TLE was saponified using 1 mL of 0.1 M potassium hydroxide (KOH) in methanol + $H_2O$ ($v/v = 9:1$) at 80 °C for 2 h. After saponification, the neutral lipids (NLs) were extracted from the TLE with $3 \times 1$ mL hexane. Hydrochloric acid (HCl; 37 %) was added to acidify the remaining TLE until pH < 2, and the fatty acids (FAs) were subsequently extracted with $3 \times 1$ mL DCM. The neutral lipids (NLs) were further separated into three polarity fractions of hydrocarbons (containing alkanes), ketones, and polar lipids through a silica column by eluting with 4 mL of hexane, DCM + hexane ($v/v = 2:1$), and DCM + methanol ($v/v = 1:1$), respectively. The fatty acids (FAs) were methylated by adding 40 μL 37 % HCl and 1 mL methanol and were heated to 50 °C for > 12 h to form fatty acid methyl esters (FAMEs). Prior to sealing, the head-space of the sample vials was filled with pure nitrogen gas. The FAME fraction was extracted from the methanol solution with $3 \times 1$ mL hexane after methylation was completed.

The hydrocarbon and FAME fractions were analysed with an Agilent 7890A gas chromatograph equipped with a flame ionization detector (GC-FID) using an Agilent J&W DB-5MS column (60 m × 250 μm × 0.25 μm thick film). The oven was held at 60 °C for 1 min, heating at a ramp of 20 °C min$^{-1}$ to 150 °C and of 6 °C min$^{-1}$ to 320 °C, with a

**Table 1.** Radiocarbon dates of cores PS51/154 and PS51/159. The calibrated ages are shown as median ages, with 95.4 % probability ranges in brackets. The ages were calibrated against the Marine20 curve (Heaton et al., 2020), with $\Delta R = -95 \pm 61$ yr.

| Lab ID | Depth (cm) | Material | Radiocarbon age ($^{14}$C yr BP) | Calibrated age (cal. yr BP) | Reference |
|---|---|---|---|---|---|
| PS51/154-11 | | | | | |
| KIA-27682 | 25 | Foraminifers/*Yoldiella* sp. | $3425 \pm 30$[a] | 3302 (3529–3044) | Taldenkova et al. (2010) |
| KIA-6919 | 31 | *Yoldiella intermedia* | $1540 \pm 45$ | | Bauch et al. (2001) |
| KIA-32811 | 39 | bivalves/gastropods | $1800 \pm 35$ | | Taldenkova et al. (2010) |
| KIA-32810 | 39 | Foraminifers | $5040 \pm 50$[a] | 5343 (5570–5065) | Taldenkova et al. (2010) |
| KIA-27683 | 51 | Foraminifers/ostracods/ *Yoldiella* sp. | $9570 \pm 60$[a] | 10286 (10530–10024) | Taldenkova et al. (2010) |
| KIA-32812 | 73 | Foraminifers | $9410 \pm 70$[a] | 10343 (10561–10117) | Taldenkova et al. (2010) |
| KIA-32813 | 73 | *Yoldiella lenticula* | $9605 \pm 45$ | | Taldenkova et al. (2010) |
| KIA-27684 | 85 | Foraminifers/*Portlandia arctica* | $9505 \pm 50$[a] | 10389 (10604–10151) | Taldenkova et al. (2010) |
| KIA-32814 | 115 | *Yoldiella lenticula* | $9630 \pm 50$[a] | 10566 (10864–10263) | Taldenkova et al. (2010) |
| KIA-32815 | 131 | *Nucula tenuis* | $10085 \pm 45$[a] | 11156 (11348–10857) | Taldenkova et al. (2010) |
| KIA-6920 | 138 | *Macoma calcarea* | $10120 \pm 55$[a] | 11182 (11383–10890) | Bauch et al. (2001) |
| KIA-6921 | 204 | *Nucula tenuis* | $10235 \pm 45$[a] | 11381 (11641–11124) | Bauch et al. (2001) |
| 11524.1.1 | 223.5 | *Macoma calcarea* | $10145 \pm 104$[a] | 11463 (11740–11170) | This study[c] |
| 11525.1.1 | 247 | *Macoma calcarea* | $10482 \pm 91$[a] | 11702 (12039–11330) | This study[c] |
| KIA-6922 | 300 | *Yoldiella intermedia* | $10725 \pm 50$[a] | 12183 (12475–11834) | Bauch et al. (2001) |
| | 339.5 | bivalves | $12040 \pm 55$[a] | 13479 (13712–13243) | Hörner et al. (2016) |
| KIA-6923 | 375 | *Yoldiella lenticula* | $12180 \pm 60$[a] | 13663 (13882–13436) | Bauch et al. (2001) |
| 11526.1.1 | 392.5 | *Macoma calcarea* | $12257 \pm 35$[a] | 13741 (13963–13514) | This study[b] |
| 11527.1.1 | 420.5 | *Macoma calcarea* | $12189 \pm 97$[a] | 13850 (14090–13582) | This study[c] |
| KIA-6924 | 440 | *Yoldiella intermedia* | $12525 \pm 55$[a] | 14091 (14417–13777) | Bauch et al. (2001) |
| KIA-6925 | 518 | Foraminifers/*Portlandia arctica* | $13120 \pm 60$[a] | 15113 (15427–14804) | Bauch et al. (2001) |
| 11529.1.1 | 520.5 | *Macoma calcarea* | $12855 \pm 122$ | | This study[c] |
| 11528.1.1 | 530.5 | *Yoldiella* sp. | $14502 \pm 44$ | | This study[b] |
| KIA-9976 | 567 | Foraminifers | $13540 \pm 90$[a] | 15659 (15971–15339) | Taldenkova et al. (2010) |
| KIA-9977 | 569 | Foraminifers | $13570 \pm 110$[a] | 15687 (16006–15365) | Taldenkova et al. (2010) |
| PS51/159-10 | | | | | |
| KIA-6927 | 11 | *Macoma* sp. | $845 \pm 30$[a] | 380 (547–175) | Bauch et al. (2001) |
| 11523.1.1 | 32.5 | *Portlandia arctica* | $2940 \pm 24$[a] | 2633 (2841–2391) | This study[b] |
| KIA-6928 | 56 | *Portlandia arctica* | $4980 \pm 35$[a] | 5194 (5446–4944) | Bauch et al. (2001) |
| | 72.5 | bivalve | $6610 \pm 40$ | | Hörner et al. (2016) |
| KIA-6929 | 90 | *Portlandia arctica* | $6305 \pm 35$[a] | 6629 (6865–6399) | Bauch et al. (2001) |
| KIA-6930 | 131 | *Portlandia arctica* | $8955 \pm 40$[a] | 9526 (9790–9305) | Bauch et al. (2001) |
| KIA-6931 | 215 | *Portlandia arctica* | $9420 \pm 50$[a] | 10160 (10416–9886) | Bauch et al. (2001) |
| KIA-6932 | 315 | *Portlandia arctica* | $9650 \pm 45$[a] | 10501 (10773–10230) | Bauch et al. (2001) |
| KIA-6933 | 410 | *Portlandia arctica* | $10720 \pm 55$[a] | 12012 (12376–11703) | Bauch et al. (2001) |
| KIA-6934 | 485 | *Portlandia arctica* | $11060 \pm 70$[a] | 12517 (12732–12210) | Bauch et al. (2001) |

[a] Dating results that were taken for the age–depth model calculation. [b] Sample analysed as graphite. [c] Sample analysed as $CO_2$ gas.

final holding time of 35 min. $n$-C$_{15-34}$ alkanes and $n$-C$_{14-32}$ fatty acids were identified by comparing the retention times of an $n$-C$_{10-40}$ alkane standard mix and an $n$-C$_{28:0}$ FAME standard, respectively. The uncertainty was calculated as the average of the absolute differences between the mean values from duplicate analyses; this was expressed as standard variation (cf. Grotheer et al., 2015). The uncertainty of the total contents was 11.6 % ($n = 62$) for $n$-alkanes and 12.8 % ($n = 68$) for FAMEs.

## 3.5    Lignin phenol extraction and analysis

The extraction procedure for lignin oxidation products was modified from the protocol described by Goñi and Montgomery (2000). About 200–300 mg of dried and homogenized samples, around 300 mg of copper oxide (CuO), around 50 mg of ferrous ammonium sulfate, and 6 mL NaOH (2 N) were added into a Teflon tube in an oxygen-free environment (O$_2$ < 1.0 %). Samples were oxidized in a microwave digestion system (MARS 6) at 150 °C for 90 min. Afterwards, 9.75 µg of ethyl vanillin (Evl) was added as an internal standard; the samples were then centrifuged, and the supernatant was recovered. The solutions were acidified with 3 mL HCl (6 N). The CuO oxidation products, including lignin phenols, were recovered by liquid/liquid extraction with 2 × 5 mL ethyl acetate. Sodium sulfate (Na$_2$SO$_4$) was added to the ethyl acetate extract to remove any remaining water. The extracts were evaporated to dryness using a rotational vacuum concentrator and were subsequently transferred into analytical vials with 500 µL pyridine for later analysis. Prior to analysis, 40 µL of the extracts dissolved in pyridine were silylated by adding 40 µL N,O-Bis(trimethylsilyl)trifluoroacetamide with 1 % trimethylchlorosilane and reacted at 50 °C for 15 min. The CuO oxidation products were analysed with an Agilent Technologies 7820A gas chromatograph coupled with a 5977B MSD mass selective detector (GC-MS) equipped with a 30 m × 320 µm × 0.25 µm thick film column (Trajan SGE PB-1). The oven was set from 95 to 300 °C at a heating ramp of 4 °C min$^{-1}$ with a holding time of 10 min. Here we focused on 3,5-dihydroxybenzoic acid (3,5 Bd) and three lignin phenol groups: (1) vanillyl phenols (V), including vanillin (Vl), acetovanillone (Vn), and vanillic acid (Vd); (2) syringyl phenols (S), including syringaldehyde (Sl), acetosyringone (Sn), and syringic acid (Sd); and (3) cinnamyl phenols (C), including $p$-coumaric acid ($p$Cd) and ferulic acid (Fd). The uncertainty of the total content for lignin phenols was 6.9 % ($n = 10$).

## 3.6    Biomarker parameters

We used the $n$-alkane *Sphagnum* proxy defined by Vonk and Gustafsson (2009) as the ratio of the $n$-C$_{25}$ alkane content to the sum of $n$-C$_{25}$+C$_{29}$ alkane contents shown in Eq. (1). Previous studies have shown that *Sphagnum* mosses, which

are abundant in peatlands across northern high-latitude regions, predominantly produce mid-chain $n$-alkanes, particularly $n$-C$_{25}$ (van Dongen et al., 2008; Vonk and Gustafsson, 2009). In contrast, higher plants primarily synthesize long-chain $n$-alkanes (also recognized as high-molecular-weight (HMW) $n$-alkanes) (Bianchi and Canuel, 2011). A higher C$_{25}$ / (C$_{25}$+C$_{29}$) value therefore reflects a greater contribution from peatland-derived sources. The uncertainty of the C$_{25}$ / (C$_{25}$+C$_{29}$) proxy was determined to be 0.006, based on the average standard variation of 31 pairs of duplicate analyses ($n = 62$).

$$Sphagnum\ \text{proxy} = \frac{C_{25}}{C_{25} + C_{29}} \tag{1}$$

[TS2] The carbon preference index (CPI) was calculated as the ratio of odd-numbered carbon $n$-alkanes to even-numbered carbon $n$-alkanes, as shown in Eq. (2) (Bray and Evans, 1961). Lower CPI values indicate greater thermal maturity of OMs, which may result from increased OM degradation (Angst et al., 2016) or contributions from petrogenic sources (Bray and Evans, 1961). The uncertainty of the CPI was 0.05 based on the average standard variation of 31 pairs of duplicate analyses ($n = 62$).

$$\text{CPI} = \frac{C_{23} + 2 \times C_{25} + 2 \times C_{27} + 2 \times C_{29} + 2 \times C_{31} + C_{33}}{2 \times (C_{24} + C_{26} + C_{28} + C_{30} + C_{32})} \tag{2}$$

We used the HMW fatty acids to calculate the temporal change in mass accumulation rate (MAR) to evaluate the terrOM flux into marine basins. Fatty acids in higher plants are dominated by even-numbered HMW saturated fatty acids (Bianchi and Canuel, 2011). The MAR of HMW fatty acids was calculated from Eq. (3). $S$ represents the sedimentation rate (cm yr$^{-1}$); $\rho$ refers to dry bulk density (g cm$^{-3}$); and $C_{24:0}$, $C_{26:0}$, $C_{28:0}$, and $C_{30:0}$ are the contents of the HMW $n$-fatty acids per gram of dry sediment (mg g$^{-1}$). [TS3] Data on the unit of MAR (mg cm$^{-2}$ kyr$^{-1}$) can be found in Tables S1 and S2 in the Supplement. The average uncertainty of the MAR of HMW fatty acids was 14.1 % ($n = 68$). One can also use other indices that represent the terrOM source. We calculated the MARs of HMW $n$-alkanes and lignin phenols in cores PS51/154 and PS51/159 as well. The patterns between all the MARs were identical (Fig. S3). Here, we only show the MAR of HMW fatty acids for a better comparison to previous studies (Winterfeld et al., 2018; Meyer et al., 2019; Queiroz Alves et al., 2024).

$$\text{MAR} = S \times \rho \times (C_{24:0} + C_{26:0} + C_{28:0} + C_{30:0}) \tag{3}$$

We used lignin phenols to calculate three indices. The vanillic acid over vanillin ratio (Vd / Vl) reflects the degree of degradation of lignin in the sediments, with higher Vd / Vl indicating more degraded lignin (Hedges et al., 1988). The uncertainty of Vd / Vl was 0.01 based on the average standard variation of five pairs of duplicate analyses ($n = 10$).

The syringyl phenols over vanillyl phenols ratio (S / V) indicates the relative contribution of gymnosperm and angiosperm sources, and the cinnamyl phenols over vanillyl phenols ratio (C / V) illustrates the relative contribution of woody and non-woody tissue (Hedges and Mann, 1979; Goñi and Hedges, 1992; Tesi et al., 2010). The combination of S / V and C / V ratios is a good indicator for identifying lignin phenol sources. The S / V ratio and C / V ratio were calculated with the lignin phenol contents, as shown in Eqs. (4) and (5). The uncertainty of S / V was 0.003 ($n = 10$), and the uncertainty of C / V is 0.01 ($n = 10$).

$$\frac{S}{V} = \frac{Sl + Sn + Sd}{Vl + Vn + Vd} \tag{4}$$

$$\frac{C}{V} = \frac{pCd + Fd}{Vl + Vn + Vd} \tag{5}$$

## 4 Results

### 4.1 Chronology and organic matter mass accumulation rates in cores PS51/154 and PS51/159

An updated age–depth model was established with 15 radiocarbon dates for core PS51/154 and with 11 radiocarbon dates for core PS51/159 (Table 1 and Figs. S1, S2). Core PS51/154 covered the period from 17.5 to 3.0 kyr BP. Three periods of peak sedimentation rates were found: during the end of Bølling–Allerød (from 14.1 to 13.2 kyr BP), during the Preboreal (from 11.6 to 10.9 kyr BP), and during the early Holocene (from 10.9 to 10.1 kyr BP) (Fig. S3). After 10.1 kyr BP, the sedimentation rate in core PS51/154 dropped drastically. Core PS51/159 covered the period from 11.9 to 0.3 kyr BP, with a peak of sedimentation rate during the early Holocene (from 10.6 to 9.5 kyr BP), followed by a significant sedimentation rate drop afterwards. The MARs of all biomarkers were largely affected by the pronounced sedimentation rate changes and thus showed similar temporal changes in all terrestrial biomarkers, including HMW $n$-alkanes, HMW fatty acids, and lignin phenols (Fig. S3; contents of each biomarker in Fig. S4). Figures 2a and 5g and h show the MAR of HMW fatty acids as a representation.

### 4.2 Bulk organic records in cores PS51/154 and PS51/159

The record before 16.2 kyr BP in core PS51/154 was characterized by high $\delta^{13}$C ($-23.8$‰) and low TOC values (0.57 %) (Fig. S5). A slump layer at 15.4 kyr BP (530–540 cm) was identified by increased grain size (Taldenkova et al., 2010) and a drop in total organic matter (TOC) (Hörner et al., 2016) (Fig. S5). However, the $\delta^{13}$C value did not show a significant change at this layer. The values of $\delta^{13}$C and TOC in cores PS51/154 ($-25.6$‰ and 0.79 %) and PS51/159 ($-25.8$‰ and 0.99 %) remained rather constant between 16.2 and 9.5 kyr BP, despite the three periods of peak MAR of terrestrial biomarkers (Fig. S5). After the early

Holocene (9.6 kyr BP), the $\delta^{13}$C value increased, accompanied by a drop in sedimentation rate in both cores, indicating a decrease in terrestrial input resulting from fast marine transgression (Bauch et al., 1999; Mueller-Lupp et al., 2000; Bauch et al., 2001).

### 4.3 Lipid and lignin phenol records in cores PS51/154 and PS51/159

The $C_{25} / (C_{25}+C_{29})$ ratio in core PS51/154 dropped to 0.27 between 17.0 and 16.0 kyr BP (Fig. 2b), remained stable afterward, began to increase after 12.6 kyr BP, and reached its highest value of 0.41 at 10.8 kyr BP. In core PS51/159, the $C_{25} / (C_{25}+C_{29})$ ratio remained high (0.38) before the early Holocene but decreased subsequently (Fig. 2b). The CPI index in core PS51/154 remained low before 16.0 kyr BP, averaging 5.3. The CPI value then increased and remained high until 13.0 kyr BP, with an average of 6.3. Afterwards, the CPI value declined, reaching a low value of 5.3 at 10.8 kyr BP, and remained low thereafter (Fig. 2c). In core PS51/159, the CPI value decreased from 6.0 during the Preboreal to 5.3 during the mid-Holocene, remained low between 5.5 and 2 kyr BP, and increased slightly to 5.6 in the late Holocene (Fig. 2c). Despite variability, the CPI values in cores PS51/154 and PS51/159 (4.5–6.5) range within the typical CPI value for surface and deep permafrost (Sánchez-García et al., 2014; Wild et al., 2022) and remain significantly higher than CPI values from petrogenic sources ($\sim 1$) (Bray and Evans, 1961).

In the record from core PS51/154, the Vd / Vl ratio was high (1.05) before 16.2 kyr BP (Fig. 2d). The Vd / Vl ratio remained stable in both cores before 10 kyr BP, of 0.68 in PS51/154 and 0.77 in PS51/159. In core PS51/159, the Vd / Vl ratio increased since 8.8 kyr BP and reached a stable high value of 1.00 after 6.0 kyr BP. The S / V ratio in core PS51/154 was low (0.64) before 16.0 kyr BP; the ratio increased afterward, peaked at 11.5 kyr BP with a ratio of 0.82, and decreased subsequently (Fig. 3). The core PS51/159 record exhibited the same trend. The S / V ratio reached its highest of 0.94 at 11.9 kyr BP, decreased afterwards, and reached a stable low value of 0.70 after 9.5 kyr BP (Fig. 3). The C / V ratio remained constant ($\sim 0.51$) in both records, except for one drop at the depth of the slump layer to 0.30 at 15.4 kyr BP in core PS51/154 (Fig. 3).

## 5 Discussion

### 5.1 Temporal changes of terrOM characteristics in the western Laptev Sea since the Last Deglaciation

The record from core PS51/154 before 16 kyr BP suggests that land-to-ocean terrOM transport was low, which could have resulted from permanent sea-ice cover (Hörner et al., 2016) (Fig. 2e) and the low terrestrial supply due to cold and dry hinterland conditions (Andreev et al., 2003). Specifically,

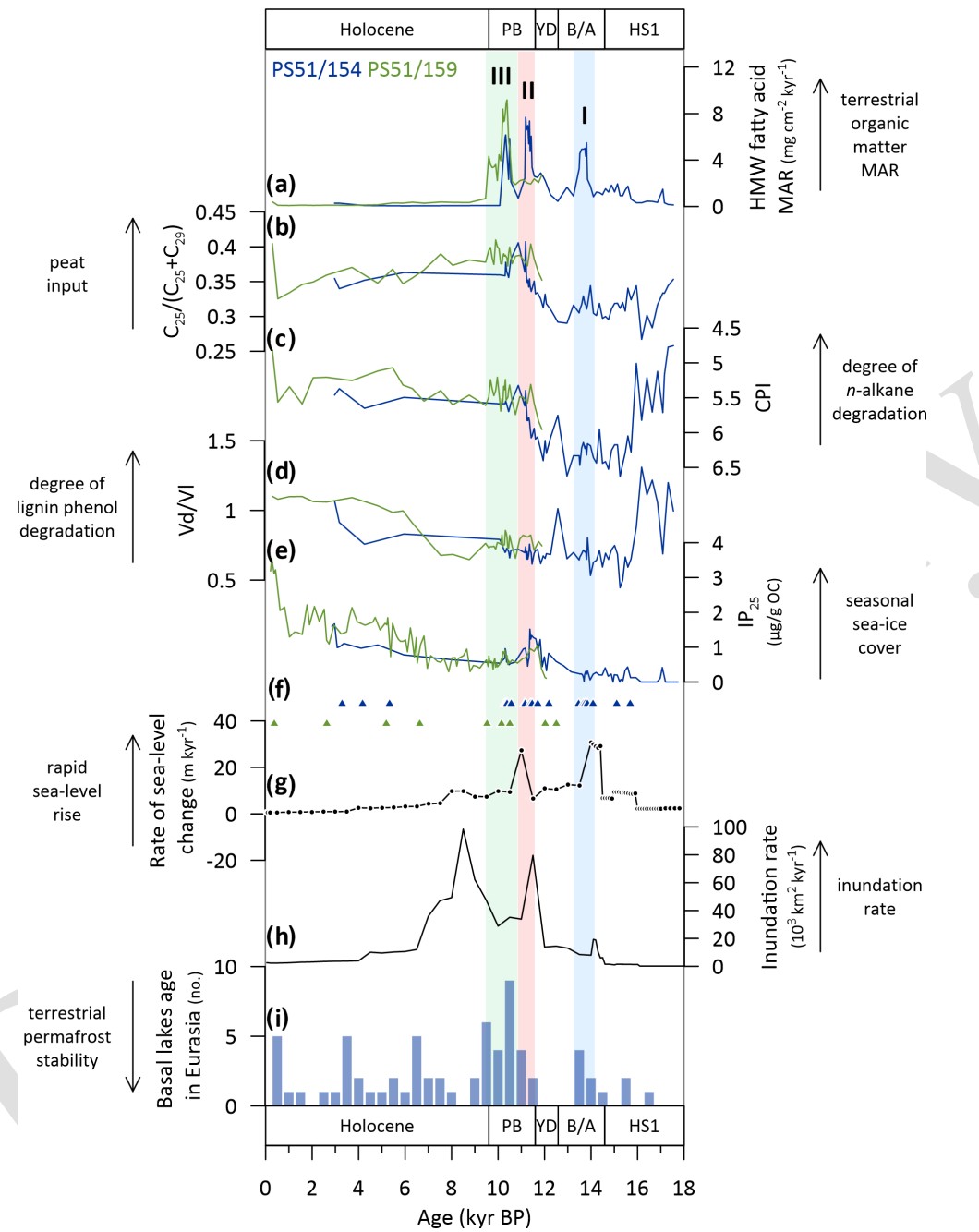

**Figure 2. (a–f)** The biomarker proxies from cores PS51/154 (dark blue) and PS51/159 (light green) and **(g–i)** environmental changes in the western Laptev Sea since the Last Deglaciation. **(a)** High-molecular-weight (HMW) fatty acid ($n$-C$_{24:0}$ + $n$-C$_{26:0}$ + $n$-C$_{28:0}$ + $n$-C$_{30:0}$) mass accumulation rate (MAR) of cores PS51/154 and PS51/159 (this study); the HMW fatty acid MAR peaks are labelled with numbers. **(b)** C$_{25}$ / (C$_{25}$+C$_{29}$) ratio of cores PS51/154 and PS51/159 (this study). **(c)** Carbon preference index (CPI) of cores PS51/154 and PS51/159 (this study). **(d)** Vanillic acid/vanillin ratio (Vd / Vl) of cores PS51/154 and PS51/159 (this study). **(e)** IP$_{25}$ contents of cores PS51/154 and PS51/159 (Hörner et al., 2016). **(f)** Age–depth model controlling points from radiocarbon dating measurements of cores PS51/154 and PS51/159. **(g)** Rate of sea-level rise in the western Laptev Sea (Klemann et al., 2015). **(h)** Area of land inundation per kyr in the western Laptev Sea, calculated from the sea-level reconstruction of Klemann et al. (2015) and the bathymetric data from the International Bathymetric Chart of the Arctic Ocean (IBCAO) (Jakobsson et al., 2012). **(i)** Counts of newly developed thermokarst lakes, categorized by the basal ages of the reported thermokarst lakes (number within 500-year bins) in Siberia (Brosius et al., 2021). The colour bars highlight the periods with HMW fatty acid MAR peaks from 14.1 to 13.2 kyr BP (blue, terrOM MAR peak I), from 11.6 to 10.9 kyr BP (red, terrOM MAR peak II), and from 10.9 to 9.5 kyr BP (green, terrOM MAR peak III). The names of different palaeoclimate periods are indicated by acronyms (HS1: Heinrich Stadial 1; B/A: Bølling–Allerød; YD: Younger Dryas; PB: Preboreal).

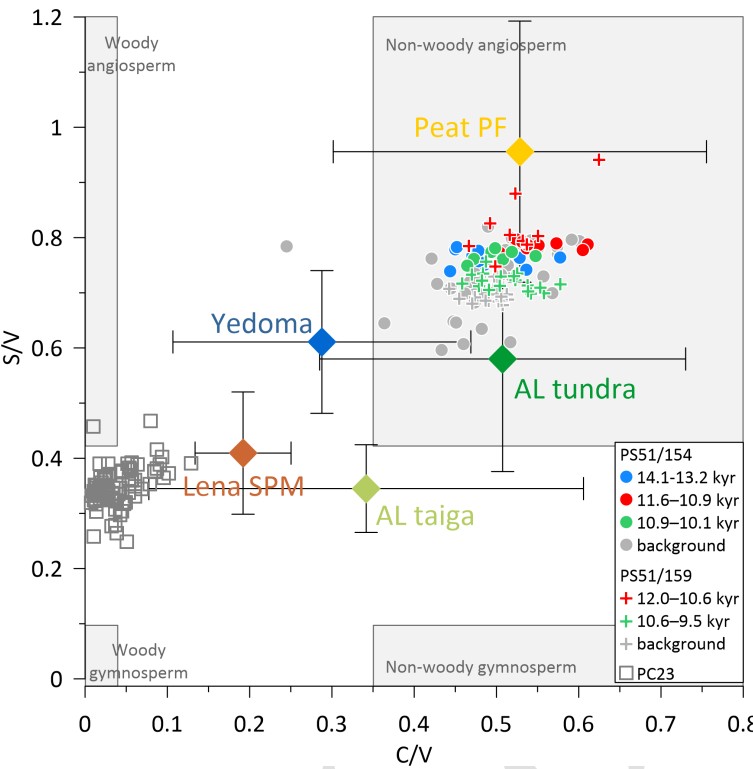

**Figure 3.** CE1 Vegetation source parameters derived from ratios of syringyl/vanillyl phenols (S / V) and cinnamyl/vanillyl phenols (C / V) of core records in the Laptev Sea and terrestrial records from Siberia. The core records include cores PS51/154 (dots; this study), PS51/159 (crosses; this study), and PC23 (Tesi et al., 2016b). For data from PS51/154 and PS51/159, symbol colours mark data from different terrOM MAR peaks: blue marks the terrOM MAR peak I period (from 14.1 to 13.2 kyr BP in core PS51/154), red marks the terrOM MAR peak II in core PS51/154 (from 11.6 to 10.9 kyr BP) and the core bottom in core PS51/159 (from 12 to 10.6 kyr BP), and green marks terrOM MAR peak III (from 10.9 to 10.1 kyr BP in core PS51/154 and 10.6 to 9.5 kyr BP in core PS51/159). Diamond data points indicate Siberian terrestrial records, including Holocene peat permafrost collected from the Lena Delta (peat PF) (Winterfeld et al., 2015), suspended particulate matter from the Lena Delta (Lena SPM) (Winterfeld et al., 2015), Yedoma (Tesi et al., 2014), the active layer from the tundra region (AL tundra) (Tesi et al., 2014), and the active layer from the taiga region (AL taiga) (Tesi et al., 2014).

high $\delta^{13}$C and low OC-normalized terrestrial biomarker contents suggest limited terrOM input from land (Fig. S5). The high Vd / Vl ratio and low CPI value during this period indicate either enhanced degradation of the small amount of terrOM reaching the core site due to longer transport times from land to the shelf caused by sea-ice blockage or, alternatively, the terrOM originated from an already degraded pool (Fig. 2c, d).

Three distinct peaks of HMW fatty acid MAR were recorded in core PS51/154, occurring from 14.1 to 13.2 kyr BP (terrOM MAR peak I), from 11.6 to 10.9 kyr BP (terrOM MAR peak II), and from 10.9 to 10.1 kyr BP (terrOM MAR peak III) (Fig. 2a). Core PS51/159 recorded one HMW fatty acid MAR peak, from 10.6 to 9.5 kyr BP (terrOM MAR peak III) (Fig. 2a). The absence of terrOM MAR peaks before 10.6 kyr BP in core PS51/159 may be because the water depth at this location was too shallow and because the main sediment depocentre of the Laptev Sea was located further north at that time (Stein and Fahl, 2000; Bauch et al., 2001). The lack of terrOM MAR peak II could also be due to the lack of available chronology tie points in core PS51/159 during this period (Fig. 2f). In addition to reflecting elevated terrestrial input, these elevated HMW fatty acid MAR peaks could also indicate enhanced organic matter preservation in marine sediments. However, disentangling these two factors is challenging, primarily because marine primary production likely increased during periods of elevated terrOM export. Terrestrial nutrients serve as a critical source that fuels marine primary production in the Arctic Ocean (Terhaar et al., 2021). As a result, increased terrOM input could stimulate marine organic matter production, leading to concurrent variations in the contents of marine-source and terrestrial-source organic matters (Fig. S6). Notably, from the Bølling–Allerød to the early Holocene, the Vd / Vl ratio remained relatively consistent, while the CPI value showed a gradual decrease (Fig. 2c, d). Neither the Vd / Vl ratio nor the CPI value exhibited noticeable differences between periods of high and low HMW fatty acid MAR. This lack of variation hinted that terrOM preservation did not significantly increase during the HMW fatty acid MAR peak periods.

Our records suggest that different mechanisms of terrOM mobilization led to varying terrOM characteristics. During the period of terrOM MAR peak I, rapid sea-level rise resulted in enhanced coastal erosion on the steep coast near the shelf break (Fig. 4a). The intruding warm Atlantic seawater inhibited sea-ice formation, further aggravating coastal erosion (Taldenkova et al., 2010; Hörner et al., 2016) (Figs. 2a, e, 4a). As observed in modern environments, coastal areas with high cliffs experience accelerated erosion when rapid sea-level rise causes waves to reach higher steep cliffs (Limber et al., 2018; Shadrick et al., 2022). The relatively cold and dry Siberian inland hindered peatland development (Fig. 2i) (Hubberten et al., 2004; Brosius et al., 2021). The combination of enhanced coastal erosion and limited peat sources is reflected in low $C_{25} / (C_{25}+C_{29})$ proxy and low S / V ratios during terrOM MAR peak I compared to the later terrOM MAR peak periods (Figs. 2b, 4a). The relatively high CPI value and low Vd / Vl ratio also implied that the terrOM was sourced from less degraded terrestrial permafrost.

During the periods of terrOM MAR peak II and terrOM MAR peak III, increasing temperature and humidity in the Siberian hinterland facilitated contemporary peat development, as reflected by the increasing tundra and shrub vegetation (Andreev et al., 2003, 2011; Hubberten et al., 2004) and peat cover (Smith et al., 2004; Yu et al., 2010) in northern Siberia. The increasing temperature also facilitates thermokarst slump formation, which exposed deep, old peatland (Schuur et al., 2015) (Fig. 2i). The peak $C_{25} / (C_{25}+C_{29})$ ratio values during this period indicate that terrestrial peat deposits, either developed contemporarily or exposed from deep permafrost, were transported by river or/and widely flooded shelf and mobilized (Fig. 2b). The elevated peat input during terrOM MAR peak II was also reflected by an increased soft angiosperm tissue contribution, originating from grass-like material abundant in Holocene peats and Holocene permafrost (Winterfeld et al., 2015) (Fig. 3). The S / V ratio decreased during terrOM MAR peak III, indicating an elevated contribution of gymnosperm plants, likely reflecting the northward expansion of the conifer tree line due to rising temperature and humidity during this period (Hubberten et al., 2004; Wild et al., 2022) (Figs. 2b, 3). During this period, the Laptev Sea shelf was rapidly inundated due to its flat topography, and the inundation rate remained high even after the sea-level rise slowed down (Mueller-Lupp et al., 2000; Bauch et al., 2001; Klemann et al., 2015) (Figs. 2g, 4b, c). In coastal areas with gentle slopes, rapid sea-level rise results in accelerated marine transgression and inundates terrestrial permafrost. Seawater also increases the temperature of the inundated permafrost and facilitates thawing (Overduin et al., 2016). The rapid inundation could be a major driver of elevated terrOM mobilization. Increased inland warming and shelf inundation rates likely facilitate terrOM degradation both on land and during cross-shelf transport (Bröder et al., 2018; Brosius et al., 2021), as evidenced by the decreasing CPI value in core

PS51/154 (Fig. 2c). However, the Vd / Vl ratio during this period remained consistent with that of the previous period of terrOM MAR peak I (Fig. 2d). Since $n$-alkanes are predominately sourced from deep permafrost, whereas lignin phenols primarily reflect surface transport (Feng et al., 2013, 2015), the divergence between these two indices may indicate a shift in terrOM source. Our records highlight that variations in terrOM reflect changes in inland permafrost stability and marine transgression dynamics.

An additional source of brassicasterol, typically ascribed to marine diatoms, during the period of terrOM MAR peak II in core PS51/154 was attributed to increased river runoff during this period (Hörner et al., 2016). This additional river runoff likely did not originate from the Lena River, as the lignin phenol assemblages in core PS51/154 differed significantly from those of the nearby sediment core records (PC23) (Tesi et al., 2016b) and Lena River suspended particulate matter (Lena SPM) (Winterfeld et al., 2015) (Fig. 3). Instead, the riverine terrOM deposited in the western Laptev Sea likely originated from the Olenyok or Khatanga rivers rather than the Lena River (Fig. 4b). This is supported by modern observations that most freshwater and suspended sediment discharged from the Lena River are transported eastward to the eastern Laptev Sea (Dmitrenko et al., 1999; Guay et al., 2001), and the mineral assemblages in western Laptev Sea surface sediments resemble those from the Khatanga River rather than the Lena River (Dethleff et al., 2000).

Overall, the S / V and C / V ratios in cores PS51/154 and PS51/159 are higher than those in the Lena SPM (Fig. 3) (Winterfeld et al., 2015). The low S / V and C / V ratios in the Lena SPM suggest a significant contribution of woody gymnosperm tissues transported from boreal forests in the southern part of the Lena River catchment (Tesi et al., 2016b; Wild et al., 2022). In contrast, the higher S / V and C / V ratios in PS51/154 and PS51/159 indicate that the vegetation source of the western Laptev Sea primarily originated from higher latitudes, reflecting a regional signal. This is further supported by the less degraded lignin phenol signals (lower Vd / Vl ratio) in cores PS51/154 and PS51/159 compared to those in the Lena SPM, which could be due to either a shorter transport distance or better-preserved organic matter that was freeze-locked in permafrost (Wild et al., 2022).

## 5.2 Environmental change affecting rapid organic matter transport to the ocean on cross-Arctic and regional scales

In order to evaluate whether the described terrOM MAR peaks reflected cross-Arctic-scale climate dynamics, we compared our HMW fatty acid MAR results with other terrOM MAR records from the Arctic marginal seas. Studies have used different biomarkers to trace terrestrial inputs, including lignin phenols, HMW $n$-alkanes, and campesterol and $\beta$-sitosterol. Campesterol and $\beta$-sitosterol are biosynthesized in vascular plants and thus reflect terrestrial sig-

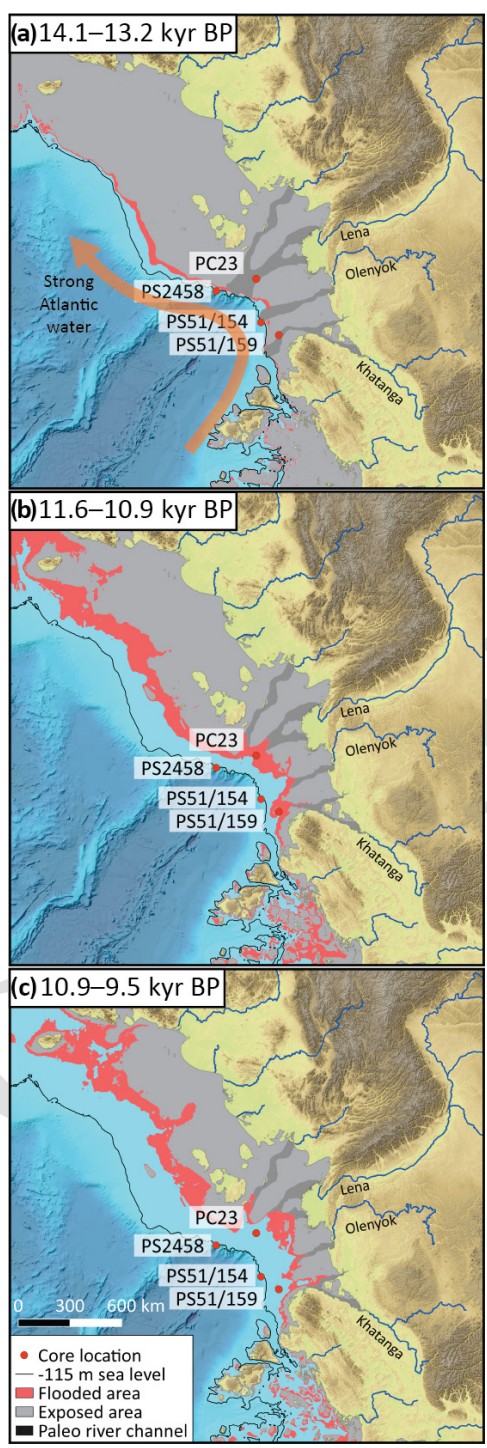

**Figure 4.** Flooded shelf area during the periods of high HMW fatty acid MAR recorded in cores PS51/154 and PS51/159: **(a)** terrOM MAR peak I, from 14.1 to 13.2 kyr BP; **(b)** terrOM MAR peak II, from 11.6 to 10.9 kyr BP; and **(c)** terrOM MAR peak III, from 10.9 to 9.5 kyr BP. The grey area shows the exposed continental shelf. The red areas show the flooded area during the periods. The flooded and exposed areas are calculated using the modern bathymetry map from IBCAO (Jakobsson et al., 2012). The river data are from Lehner and Grill (2013). The dark-grey areas denote the palaeo-river channel (Kleiber and Niessen, 2000). The thick black line denotes the contour of the −115 m sea level, which approximates the sea level at 18 kyr BP. Red dots are the core locations in the Laptev Sea, including cores PC23 (Tesi et al., 2016b), PS2458 (Spielhagen et al., 2005), PS51/154 (this study), and PS51/159 (this study). The orange arrow indicates the period that the Laptev Sea shelf experienced a strong impact from Atlantic water (Taldenkova et al., 2010). The maps are plotted with QGIS3.22.9 (QGIS Geographic Information System; QGIS Association).

nals, in contrast to dinosterol and brassicasterol, which are found abundantly in marine plankton (Bianchi and Canuel, 2011). Data were collected from sites in the Beaufort Sea (ARA04C/37) (Wu et al., 2020), the Laptev Sea (PC23) (Tesi et al., 2016b), the northern Svalbard continental margin (HH11-09) (Nogarotto et al., 2023), and the Fram Strait (PS2837-5 and MSM05/5-712-2) (Birgel and Hass, 2004; Aagaard-Sørensen et al., 2014; Müller and Stein, 2014; Zamelczyk et al., 2014) (Figs. 1, 5).

TerrOM MAR largely depends on changes in sedimentation rate (Figs. 5, S7), which can vary significantly between age models. Using age models with dense chronological control points is crucial. To ensure confidence in the timing of terrOM MAR peaks, we selected records with multiple chronological tie points both below and above the identified terrOM MAR peaks (Fig. 5d, f, g, h) and multiple records from the same region to provide a more comprehensive regional pattern (Fig. 5f–h, i–k).

Age–depth models for these records were recalibrated against the Marine20 calibration curve (Heaton et al., 2020) or a combination of the IntCal20 (Reimer et al., 2020) and Marine20 curves, depending on the original studies. For cores PC23 and HH11-09, which already have published age models updated to the Marine20 or Marine20 + IntCal20 curves, we adopted the existing age models (Nogarotto et al., 2023; Sabino et al., 2024). For records from the Fram Strait (PS2837-5, MSM05/5-712-2), updated $\Delta R$ values were derived from the Marine20 database. We applied a constant updated $\Delta R$ for calibration, following their previously published age models (Birgel and Hass, 2004; Aagaard-Sørensen et al., 2014; Müller and Stein, 2014; Zamelczyk et al., 2014). Since no $\Delta R$ values were available for the Beaufort Sea region in the Marine20 database, new $\Delta R$ values for each dating point were calculated by subtracting 150 years from the previous $\Delta R$ values for each point (Keigwin et al., 2018), following the guidelines in Heaton et al. (2023). Detailed information on the updated $\Delta R$ values is provided in Table S3. We further calculated the age uncertainty of each terrOM MAR peak by including ± 1$\sigma$ uncertainty from the age models. The possible age ranges of these terrOM MAR peaks are shown in Fig. 5.

### 5.2.1 Effects of global sea-level rise

The rapid global sea-level rise during meltwater pulse 1A (mwp-1A) was an important process in terrOM mobilization across different regions in the Arctic. TerrOM MAR peaks during this period are observed widely in records from the Eurasian Arctic (Fig. 5b, d, g, i–k) (Birgel and Hass, 2004; Lambeck et al., 2014; Nogarotto et al., 2023). Inland temperatures in both North America and Siberia remained low during this time (Fig. 5c, e). These concurrent terrOM MAR peaks suggest that rapid sea-level rise was the primary cause for circumarctic permafrost mobilization, possibly contributing to the rapid increase in atmospheric $CO_2$ concentration

(Marcott et al., 2014; Winterfeld et al., 2018). The record from core ARA04C/37 did extend to the mwp-1A period. However, increased transport of old terrOM was observed at the terrOM MAR peak in ARA04C/37 during the period between 13.6 and 13 kyr BP, suggesting contributions from deep permafrost mobilized from coastal erosion (Wu et al., 2022). This terrOM peak may have begun earlier than the time period covered by the record and is likely associated with the rapid sea-level rise during the mwp-1A period (Fig. 5b).

During mwp-1B, terrOM MAR also increased in records from the Beaufort Sea and the Laptev Sea (Fig. 5b, d, f, g). However, this peak was absent in records from Svalbard and the Fram Strait (Fig. 5i, j, k). One of the possible reasons could be the lack of radiocarbon age controls in these cores during this period. Also, the amplitude of sea-level rise and even the existence of mwp-1B remain debated (Lambeck et al., 2014). The lack of widespread terrOM MAR during mwp-1B suggests that this event might not affect the pan-Arctic region as extensively as mwp-1A did.

### 5.2.2 Regional processes: inland warming, sea-ice loss, and freshwater pulses

Outside the periods of mwp-1A and mwp-1B (indicated by blue bars in Fig. 5), several regional terrOM MAR peaks were observed in the Arctic marginal seas. These peaks were likely driven by regional factors such as inland warming, decreased sea-ice cover, and freshwater pulses. Warming facilitates permafrost thawing and thermokarst lake development (Schuur et al., 2015). Lake basal ages indicate the time of lake formation, and a higher count of thermokarst lake formation in a certain period thus implies intense inland warming (Brosius et al., 2021). In the Canadian Arctic, terrOM MAR peaks appeared during the interval between mwp-1A and mwp-1B (Fig. 5d). Inland warming in North America began at approximately 13.5 kyr BP, while Siberia remained cold (Brosius et al., 2021). TerrOM MAR peaks appeared during the early Holocene across the Siberian Arctic and Fram Strait records (Fig. 5g, h, k). During the early Holocene, enhanced warming in Siberian hinterlands led to elevated terrOM mobilization (Fig. 5e). These terrOM MAR peaks occurred during periods of little sea-level variation, indicating that intensified inland warming could destabilize permafrost and result in regional terrOM mobilization even in the absence of rapid sea-level rise.

Notably, the observed terrOM MAR peaks in the Arctic marginal seas coincided with regionally reduced seasonal sea-ice cover, as indicated by reduced $IP_{25}$ (Müller et al., 2009; Hörner et al., 2016; Wu et al., 2020). $IP_{25}$ refers to an alkane that is produced by sea-ice-associated diatoms (Volkman et al., 1994). The existence of $IP_{25}$ indicates the appearance of seasonal sea ice, while the lack of $IP_{25}$ implies either the lack of seasonal or permanent sea-ice coverage in the area (Belt and Müller, 2013). Studies have shown that ac-

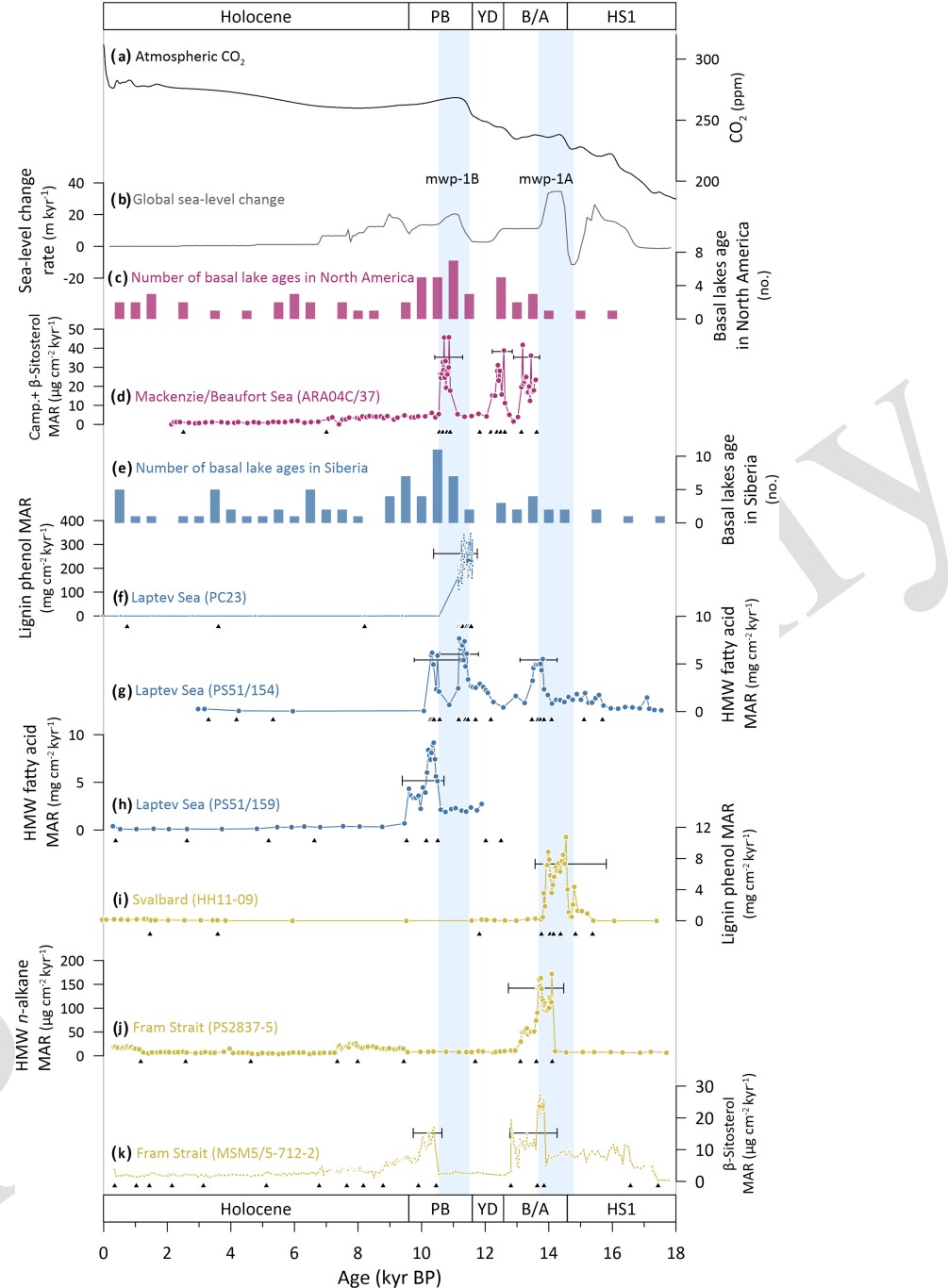

**Figure 5.** Environmental changes since the Last Deglaciation and terrestrial biomarker mass accumulation rates (MARs) of core records in the Arctic Ocean and higher-latitude Northern Hemisphere. **(a)** Atmospheric $CO_2$ concentration (Marcott et al., 2014; Köhler et al., 2017). **(b)** Rate of global sea-level change (Lambeck et al., 2014). **(c)** Compilation of basal ages of thermokarst lakes (number within 500-year bins) in North America (Brosius et al., 2021). **(d)** Campesreol $+\beta$-sitosterol MAR from core ARA04C/37, Beaufort Sea (Wu et al., 2020). **(e)** Compilation of basal ages of thermokarst lakes (number within 500-year bins) in Siberia (Brosius et al., 2021). **(f)** Lignin phenol MAR from core PC23, Laptev Sea (Tesi et al., 2016b). **(g)** HMW fatty acid MAR from core PS51/154, Laptev Sea (this study). **(h)** HMW fatty acid MAR from core PS51/159, Laptev Sea (this study). **(i)** Lignin phenol MAR from core HH11-09, northern Svalbard continental margin (Nogarotto et al., 2023). **(j)** HMW $n$-alkane MAR from core PS2837-5, Fram Strait (Birgel and Hass, 2004). **(k)** $\beta$-sitosterol MAR from core MSM05/5-712-2, Fram Strait (Aagaard-Sørensen et al., 2014; Müller and Stein, 2014; Zamelczyk et al., 2014). The blue bars highlight the period of rapid sea-level rise. The black intervals under each terrOM MAR peak indicate the age uncertainty range of the terrOM MAR peaks, calculated by the $\pm 1\sigma$ age models. Meltwater pulses are denoted as mwp-1A and mwp-1B. The names of different palaeoclimate periods are indicated by acronyms (HS1: Heinrich Stadial 1; B/A: Bølling–Allerød; YD: Younger Dryas; PB: Preboreal).

celerated coastal erosion can be mitigated by the presence of land-fast sea ice, which prevents thermal denudation by seawater, direct wave erosion, and saltwater intrusion into permafrost coasts (Rachold et al., 2000; Overduin et al., 2016; Nielsen et al., 2020; Irrgang et al., 2022). Modern observations have associated elevated coastal erosion rates in the Laptev Sea with reduced winter sea-ice cover (Nielsen et al., 2020). Modelling studies also indicate a positive correlation between inland permafrost stability and winter sea-ice extent under modern conditions (Vandenberghe et al., 2012). The terrOM MAR peak III in core PS51/154 and PS51/159 reflected that, even without rapid sea-level rise, the mobilized material caused by inland warming was more easily transported to marine basins because of the lack of sea-ice protection on the coast (Fig. 2a, e). A similar relation was shown in the core from the Beaufort Sea (ARA04C/37), as the two terrOM MAR peaks registered in the core both happened during an ice-free period (Wu et al., 2020). The Holocene record of core PS51/159 shows opposite conditions. The seasonal sea-ice cover started to increase from 7 kyr BP (Hörner et al., 2016). Even though several warming periods in Siberia were recorded since then, no terrOM MAR peak was recorded in core PS51/159 (Fig. 2a, e, i). The growing sea ice might play a role in protecting the warming land from being eroded and keeping the terrestrial material from being transported into the Laptev Sea shelf.

Freshwater floods triggered by ice-dammed lake breakings could be another regional factor in terrOM mobilization. These flooding events can be recorded by the decreasing stable oxygen isotope ratio ($\delta^{18}$O) in planktic foraminifera, *Neogloboquadrina pachyderma* (Spielhagen et al., 2005; Keigwin et al., 2018). In the Canadian Arctic record (ARA04C/37), the terrOM MAR peak occurring during the Younger Dryas was linked to a well-described meltwater flood event through the Mackenzie River, based on the drop in both $\delta^{18}$O value in *Neogloboquadrina pachyderma* and TOC (Keigwin et al., 2018; Klotsko et al., 2019; Wu et al., 2020) (Fig. S8). However, freshwater events were less likely to be the cause for terrOM MAR in the western Laptev Sea. While freshwater flooding events were recorded in an ice-dammed lake upstream of the Lena River ($14.9 \pm 2.0$ kyr BP) and in a sediment record from the Laptev Sea (PS2458, at 12 kyr BP) (Spielhagen et al., 2005; Margold et al., 2018), the timing of these events did not correspond with any of the terrOM MAR peaks in cores PS51/154 and PS51/159 (Fig. S8) or cause any grain size change in cores PS51/154 and PS51/159 (Taldenkova et al., 2010). This temporal mismatch suggests that Siberian freshwater pulses had little impact on the increase in terrestrial biomarker MAR in the western Laptev Sea.

## 6 Summary and conclusions

This study explores temporal changes in the composition and rate of terrOM mobilization to the western Laptev Sea, as indicated by lipid and lignin phenol records. Three rapid terrigenous organic matter supply events were identified from 14.1 to 13.2, from 11.6 to 10.9, and from 10.9 to 9.5 kyr BP. Each peak showed different compositional characteristics, suggesting distinct terrOM sources derived from different mechanisms. The first terrOM MAR peak was likely driven by enhanced coastal erosion, while the latter two peaks were associated with inland warming and rapid shelf inundation. The source shift was characterized by increased peat input, as evidenced by both $C_{25} / (C_{25}+C_{29})$ proxy and S / V ratios. Comparing our findings with records from across the Arctic indicates that the enhanced terrOM mobilization during the mwp-1A period was primarily driven by enhanced coastal erosion resulting from rapid global sea-level rise. However, terrOM MAR peaks did not always align with periods of rapid sea-level rise, suggesting that other regional factors, such as inland warming, lack of sea-ice protection, and freshwater floods, also played significant roles.

Overall, the study highlights that the topography of the western Laptev Sea shelf strongly influenced erosion scenarios linked to sea-level rise, leading to different terrOM sources mobilized from land to ocean. Our results suggest that, while rapid sea-level rise contributed to elevated terrOM mobilization on a cross-Arctic scale, regional factors such as inland warming, freshwater floods, and sea-ice cover decrease were responsible for regional terrOM mobilization.

According to the IPCC report, the projected global sea-level rise by the end of the 21st century could reach up to $8 \, \mathrm{m \, kyr^{-1}}$ (Church et al., 2013). While this projected rate is significantly slower than the rapid sea-level rise during the mwp-1A ($\sim 35 \, \mathrm{m \, kyr^{-1}}$) and mwp-1B ($\sim 20 \, \mathrm{m \, kyr^{-1}}$) periods (Lambeck et al., 2014), our results from the Last Deglaciation indicate that, even without such rapid sea-level rise, the combination of reduced sea-ice protection and increased hinterland warming can still mobilize substantial terrOM through regional coastal erosion. Given that the Arctic is experiencing rapid warming compared to the global average and that the destabilization of permafrost and sea-ice decrease is already underway (Meredith et al., 2019), an increase in terrOM input from coastal erosion is likely if the warming persists.

**Data availability.** Data generated in this study will be uploaded to the PANGAEA database. TS4

**Supplement.** The supplement related to this article is available online at [the link will be implemented upon publication].

**Author contributions.** GM designed the study; TWL, JH, HG, JW, and AN performed the measurements; TWL, TT, JH, HG, FA, and GM analysed the data; FA and JW verified the age models; TWL wrote the article with support from TT and GM; and all co-authors reviewed and edited the article.

**Competing interests.** The contact author has declared that none of the authors has any competing interests.

**Disclaimer.** Publisher's note: Copernicus Publications remains neutral with regard to jurisdictional claims made in the text, published maps, institutional affiliations, or any other geographical representation in this paper. While Copernicus Publications makes every effort to include appropriate place names, the final responsibility lies with the authors.

**Acknowledgements.** This study was supported by the German–Italian partnership project between the Alfred Wegener Institute and CNR-ISP on Chronologies for Polar Paleoclimate Archives (PAIGE), funded by the Helmholtz Association (grant no. PIE-0018). Florian Adolphi was supported through the Helmholtz Association (VH-NG 1501). We thank the captain, the chief scientist, the crew, and the scientific party of the *Polarstern* Expedition ARK-XIV/1b (PS51 Transdrift-V) for providing the studied material. We thank Kirsten Fahl and Rüdiger Stein for offering coring information. We thank the MICADAS radiocarbon laboratory group members at the Alfred Wegener Institute, namely Elizabeth Bonk, Torben Gentz, and Lea Philips. We appreciate help from Arnaud Nicolas with core subsampling. Katarzyna Zamelczyk and Steffen Aagaard Sørensen are acknowledged for the dry bulk density data of core MSM05/5-712-2. We appreciate the constructive suggestions from the editor, Bjørg Risebrobakken, and two anonymous reviewers. ChatGPT (version GPT-4) was used in an earlier draft of this article to edit grammar and improve sentence fluency.

**Financial support.** This research has been supported by the Helmholtz Association (grant no. PIE-0018).

The article processing charges for this open-access publication were covered by the University of Bremen.

**Review statement.** This paper was edited by Bjørg Risebrobakken and reviewed by two anonymous referees.

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

## Remarks from the language copy-editor

CE1    Sorry, I believe the new figure you sent is the same as the current one, i.e., still kyr and not kyr BP.

## Remarks from the typesetter

TS1    Please provide your updated supplement as a *.pdf file.

TS2    All journal papers typeset by Copernicus Publications follow the math typesetting regulations given by the IUPAC "Green Book" (Quantities, Units and Symbols in Physical Chemistry, 2nd Edn., Blackwell Science, available at: http://old.iupac.org/publications/books/gbook/green_book_2ed.pdf, 1993.). Chemical substances are written in roman font, as well as abbreviations (from 2 letters). That its why we could only write S, V, C in italic font. Please clarify.

TS3    Your comment was incomplete.

TS4    Please update. Note that this cannot be updated after publication.