# Peer review of "Environmental controls of rapid terrestrial organic matter mobilization to the western Laptev Sea since the last deglaciation"

_Climate of the Past, 2024_

## Author Comment (AC1)

**Response to comments by reviewer 1 on cp-2024-60**

We would like to thank Reviewer 1 for the meticulous review and valuable suggestions, which helped us to improve this manuscript significantly. Below, we address each of Reviewer 1's comments in detail. The reviewer's comments are in blue, and our responses are in black.

In this manuscript, Lin et al., evaluate changes in terrestrial organic matter input into the Arctic ocean over the last 18,000 years. The chosen study sites drain vast quantities of permafrost, such that enhanced terrestrial OC input might indicate enhanced permafrost thaw. The authors identify several pulses of terrestrial OC burial, some of which may correlate to known climatic events (e.g, meltwater pulse 1a). There is remarkably low terrOC input during the Holocene but lots of variability in terrOC delivery between 10 to 16 kyr. Each peak shows different compositional characteristics, suggesting distinct terrOM sources derived from different mechanisms. However, it would be interesting to try and unravel this further using your existing data (see comments below). Overall, the manuscript is well written, the figures are clear and the captions offer an appropriate level of detail. A few comments are included below that would be worth exploring further...

Comment 1: it is hard to tell whether the pulse in terrestrial OC is due to permafrost thaw or enhanced delivery of plant/soil OC. This could be assessed by measuring 14C values in different lipids, but I realize is beyond the scope of this paper. However, there is existing literature that could be helpful - for example, Feng et al. (2013; PNAS) explored how the 14C signature of different compound classes varied across the Pan-Arctic region and lignin phenols appear to mostly derive from recent carbon, whereas n-alkanes are derived from older carbon sources (e.g,. permafrost). Although you don't have 14C measurements, it could imply distinct sources for lignin vs n-alkanes in your samples.
...but another way to tackle this could be to look at other n-alkane indices such as the carbon preference index. This is frequently used to assess changes in OC maturity and may provide further insights into the type of terrOC that is being delivered into the marine realm, especially during the three pulses. If it was older OC, it may yield slightly lower CPI values. It may also tell you whether you are reworking old petrogenic OC into the marine realm too (which could be a CO2 source if it was oxidised; see work by Sparkes 2016 the Cryosphere, but also work by Bob Hilton/Valier Galy etc)

We appreciate Reviewer 1's suggestions. The CPI values in cores PS51/154 and PS51/159 (4.5–6.5) fall within the range typical for surface and deep permafrost (Wild et al., 2022; Sánchez-García et al., 2014), and remain significantly higher than CPI values from petrogenic sources (~1) (Bray and Evans, 1961). Degradation indices, such as CPI and the Sd/Sl ratio, show only small variations during the peaks of terrOM accumulation (Figure A1-1; Figure 2 in manuscript). Instead, CPI ratios suggest that fresher OM was generally deposited during the deglaciation between around 16 and 11 kyr BP, likely due to shorter transport distances to the core site at lower sea levels. Lignin phenol $\Sigma8$/ HMW fatty acid ratios are consistent with CPI trends, but the impact of the slumping layer is more pronounced (Figure A1-1). We will include CPI data in the supplementary plots and briefly discuss how transport distance to the core site is reflected in the biomarker data.

[Figure]

Figure A1-1. Mass accumulation rates (MARs) of high molecular weight (HMW) fatty acids, ratio of lignin phenol Σ8 to HMW fatty acids contents, and the carbon preference index (CPI) in cores PS51/154 and PS51/159. Colored bars highlight periods of HMW fatty acid MAR peaks: from 14.1 to 13.2 kyr BP (blue, MAR peak I), from 11.6 to 10.9 kyr BP (red, MAR peak II), and from 10.9 to 9.5 kyr BP (green, MAR peak III).

**Comment 2: Export vs preservation**

It is important to confirm that the increase in leaf wax mass accumulation rates is not due to enhanced preservation but is reflecting enhanced terrOC export. This could be explored by calculating MARs of of short-chain alkanes (algal-derived), mid-chain alkanes (moss or macrophyte derived), and long-chain alkanes (vascular plant) n-alkanes during the three "pulses". If all three increase, it might suggest that OC preservation is the main driver. But if only the mid- and long-chain alkane MARs increase, it would imply enhanced terrOC input.

We thank Reviewer 1's insight. However, disentangling the cause of elevated terrOM MAR peaks, whether due to enhanced terrOM export or enhanced preservation, is challenging. This difficulty arises mainly because marine primary production likely increased during periods of elevated terrOM export. Terrestrial nutrients serve as an important source to fuel marine primary production in the Arctic Ocean (Terhaar et al., 2021), and elevated terrOM export could stimulate marine primary production, leading to simultaneous increases in terrestrial and marine OM. Secondly, terrOM MAR peaks in cores PS51/154 and PS51/159 coincided with periods of low seasonal sea ice cover (Hörner et al., 2016) (Figure 2 in the manuscript). Reduced sea ice cover favors phytoplankton blooms, while extended open-water conditions provide more area and time for phytoplankton growth (Shiozaki et al., 2022), further increasing marine primary production. The contents per gram of OC (Figure A1-2) and MARs (Figure A1-3) of short-chain, mid-chain, and long-chain *n*-alkanes show no significant differences among the three homologues.

[Figure]

Figure A1-2. Contents of short-chain, mid-chain, and long-chain (HMW) *n*-alkanes in cores PS51/154 (dark blue) and PS51/159 (light green). Colored bars highlight periods of HMW fatty acid MAR peaks: from 14.1 to 13.2 kyr BP (blue), from 11.6 to 10.9 kyr BP (red), and from 10.9 to 9.5 kyr BP (green).

[Figure]

Figure A1-3. Mass accumulation rates (MARs) of short-chain, mid-chain, and long-chain (HMW) *n*-alkanes in cores PS51/154 (dark blue) and PS51/159 (light green). Colored bars highlight periods of HMW fatty acid MAR peaks: from 14.1 to 13.2 kyr BP (blue), from 11.6 to 10.9 kyr BP (red), and from 10.9 to 9.5 kyr BP (green).

Comment 3: a note on pAq
The authors use the pAq index (the ratio between mid vs long chain n-alkanes) to infer changes in wetland input, but I would note that's its more complex than this. For example, both Sphagnum moss and aquatic

macrophytes are characterized by similar lipid distributions (Baas et al., 2000; Ficken et al., 2000), so without knowledge of the local vegetation, its challenging to say whether the pAq ratio is due to changes in moss input or macrophyte input. Perhaps you could narrow this down by drawing upon predicted vegetation patterns during the Holocene/LGM etc. There is also great work by Jorien Vonk, Bart van Dongen, Orjan Gustafsson etc in similar pan-arctic regions that might be useful. For example, Vonk 2009 paper in Org. Geochem shows that "…the C25/(C25 + C29) n-alkane ratio is most suitable for terrestrial OM source apportionment in these coastal regions". This might be worth exploring alongside the pAq (although I suspect you will get similar results!).

We thank Reviewer 1's suggestion and for kindly pointing us to the relevant references. For the first part of the suggestion, we will include references on predicted moss cover in Siberia during the last deglaciation (Andreev et al., 2003; Hubberten et al., 2004) and in recent times (Van Dongen et al., 2008). For the second part of the suggestion, we compared the $C_{25}/(C_{25}+C_{29})$ *n*-alkane ratio with $P_{aq}$ (Figure A1-4). While both indices yield similar results, the $C_{25}/(C_{25}+C_{29})$ *n*-alkane ratio better reflects moss/peatland input and is less influenced by species and temperature variations (Vonk and Gustafsson, 2009; Van Dongen et al., 2008). Accordingly, we will adopt the $C_{25}/(C_{25}+C_{29})$ *n*-alkane ratio in the revised manuscript, retaining the same interpretations.

[Figure]

Figure A1-4. $P_{aq}$ (black lines) and $C_{25}/(C_{25}+C_{29})$ (blue lines) indices for cores PS51/154 and PS51/159 (this study).

Minor comments:
L63: v/v – the v's should be italicised – change throughout

We will correct this in the revised manuscript.

L205: correct that its used for macrophytes – but also for sphagnum mosses.

We appreciate reviewer 1 for pointing this out, in the revised manuscript we will replace $P_{aq}$ index with the $C_{25}/(C_{25}+C_{29})$ index and replace this sentence with an introduction of the $C_{25}/(C_{25}+C_{29})$ index.

During 16.2 and 9.5 kyr BP, both $\delta^{13}$C and TOC in cores PS51/154 and PS51/159 show no correlation with HMW fatty acid MAR, as evidenced by low R and high *p*-values (Figure A1-5). To improve clarity, we will revise the sentence of L252 as follows: "The values of $\delta^{13}$C and TOC in cores PS51/154 (−25.6‰ and 0.79%) and PS51/159 (−25.8‰ and 0.99%) remained rather constant between 16.2 and 9.5 kyr BP, despite the three periods of peak MAR of terrestrial biomarkers (Fig S5).".

[Figure]

Figure A1-5. TOC and $\delta^{13}$C versus high molecular weight (HMW) fatty acid mass accumulation rate (MAR) in cores PS51/154 and PS51/159. Data points from 16.2 to 9.5 kyr BP are highlighted and the linear regression lines are shown. The low correlation and high *p*-value indicate no significant changes in TOC and $\delta^{13}$C values did not change during this period despite elevated terrestrial organic matter inputs.

Please refer to the next reply.

We appreciate the reviewer's meticulous review. The original sentence from L365 and 366 is "Branched Glycerol Dialkyl Glycerol Tetraethers (brGDGTs) are found in membrane lipids from soil bacteria." We agree that brGDGTs are also found in aquatic and marine environments. This original sentence in L365-366 will be removed, as the brGDGT MAR record of cores SO202-18-3/6 (Meyer et al., 2019) is not from the Arctic Ocean and may not reflect comparable environmental conditions.

We will correct the typo in the revised manuscript.

**References**

Andreev, A. A., Tarasov, P. E., Siegert, C., Ebel, T., Klimanov, V. A., Melles, M., Bobrov, A. A., Dereviagin, A. Y., Lubinski, D. J., and Hubberten, H. W.: Late Pleistocene and Holocene vegetation and climate on the northern Taymyr Peninsula, Arctic Russia, Boreas, 32, 484-505, 2003.

Bray, E. and Evans, E.: Distribution of n-paraffins as a clue to recognition of source beds, Geochimica et Cosmochimica Acta, 22, 2-15, 1961.

Hörner, T., Stein, R., Fahl, K., and Birgel, D.: Post-glacial variability of sea ice cover, river run-off and biological production in the western Laptev Sea (Arctic Ocean)–A high-resolution biomarker study, Quaternary Science Reviews, 143, 133-149, 2016.

Hubberten, H. W., Andreev, A., Astakhov, V. I., Demidov, I., Dowdeswell, J. A., Henriksen, M., Hjort, C., Houmark-Nielsen, M., Jakobsson, M., Kuzmina, S., Larsen, E., Lunkka, J. P., Lysa, A., Mangerud, J., Möller, P., Saarnisto, M., Schirrmeister, L., Sher, A. V., Siegert, C., Siegert, M. J., and Svendsen, J. I.: The periglacial climate and environment in northern Eurasia during the Last Glaciation, Quaternary Science Reviews, 23, 1333-1357, 10.1016/j.quascirev.2003.12.012, 2004.

Meyer, V. D., Hefter, J., Köhler, P., Tiedemann, R., Gersonde, R., Wacker, L., and Mollenhauer, G.: Permafrost-carbon mobilization in Beringia caused by deglacial meltwater runoff, sea-level rise and warming, Environmental Research Letters, 14, 10.1088/1748-9326/ab2653, 2019.

Sánchez-García, L., Vonk, J. E., Charkin, A. N., Kosmach, D., Dudarev, O. V., Semiletov, I. P., and Gustafsson, Ö.: Characterisation of Three Regimes of Collapsing Arctic Ice Complex Deposits on the SE Laptev Sea Coast using Biomarkers and Dual Carbon Isotopes, Permafrost and Periglacial Processes, 25, 172-183, 10.1002/ppp.1815, 2014.

Shiozaki, T., Fujiwara, A., Sugie, K., Nishino, S., Makabe, A., and Harada, N.: Bottom-associated phytoplankton bloom and its expansion in the Arctic Ocean, Global Change Biology, 28, 7286-7295, 10.1111/gcb.16421, 2022.

Terhaar, J., Lauerwald, R., Regnier, P., Gruber, N., and Bopp, L.: Around one third of current Arctic Ocean primary production sustained by rivers and coastal erosion, Nature Communications, 12, 169, 10.1038/s41467-020-20470-z, 2021.

van Dongen, B. E., Semiletov, I., Weijers, J. W. H., and Gustafsson, Ö.: Contrasting lipid biomarker composition of terrestrial organic matter exported from across the Eurasian Arctic by the five great Russian Arctic rivers, Global Biogeochemical Cycles, 22, 10.1029/2007gb002974, 2008.

Vonk, J. E. and Gustafsson, Ö.: Calibrating n-alkane Sphagnum proxies in sub-Arctic Scandinavia, Organic Geochemistry, 40, 1085-1090, 10.1016/j.orggeochem.2009.07.002, 2009.

Wild, B., Shakhova, N., Dudarev, O., Ruban, A., Kosmach, D., Tumskoy, V., Tesi, T., Grimm, H., Nybom, I., Matsubara, F., Alexanderson, H., Jakobsson, M., Mazurov, A., Semiletov, I., and Gustafsson, O.: Organic matter composition and greenhouse gas production of thawing subsea permafrost in the Laptev Sea, Nature Communications, 13, 5057, 10.1038/s41467-022-32696-0, 2022.

---

## Author Comment (AC2)

**Response to comments by reviewer 2 on cp-2024-60**

We would like to thank Reviewer 2 for the constructive suggestions, which helped us improve the quality of this manuscript. Below, the reviewer's comments are presented in blue, and our responses to each point are in black.

Summary
This paper presents organic geochemical proxies used to distinguish the source of organic matter (OM) in two 17-18 kyr records from the Laptev sea. The data is integrated with pan-Arctic records in an attempt to investigate local versus global drivers of enhanced OM flux during deglaciation. It is a very nice study and paper that fits well within the scope of Climate of the Past. However, I also think there is some room for improvement in he presentation and discussion of the data. Specifically, 1) a clearer presentation of sedimentation rate changes in the studied cores (and how they impact the mass accumulation rate estimates) is important, 2) a more in-depth discussion about the timing and link to OM flux/source changes associated with previously identified meltwater events in the Laptev Sea between MWP-1A and MWP1B. These are discussed further in the points below:

Specific comments
Line 25: "Additional terrOM MAR peaks coincided with periods of enhanced inland warming, prolonged ice-free conditions, and freshwater flooding, which varied between regions.". I wonder if this sentence could be more specific, for instance, be specific about what periods evidence for coastal erosion in response to sea-level rise are identified, and at what periods are inland warming and freshwater flooding seen? Also - maybe the last sentence could be re-written so that the expression 'regional terrOM fluxes' only occurs once.

We thank Reviewer 2's suggestion to enhance the clarity of the abstract. We will revise the last section of the abstract (line 25) as follows: "Additional terrOM MAR peaks varied regionally. Peaks from the Beaufort Sea during the Bølling-Allerød coincided with a freshwater flooding event, while peaks from the Laptev Sea and the Fram Strait during the Preboreal/early Holocene coincided with periods of enhanced inland warming and prolonged ice-free conditions. Our results highlight the influence of regional environmental conditions, in addition to global drivers, which can either promote or preclude regional terrOM fluxes."

Could the authors add a bit more information on how a dR value of -95 +/- 65 years for the Marine20 calibration curve was derived from Bauch et al, 2001? It is good to describe the conversion of old dR values when making them compatible with the Marine20 calibration curve, it is good bookkeeping.

We thank Reviewer 2's suggestion to provide more details about the updated ΔR values for cores PS51/154 and PS51/159, this helps improving the clarity of the calibration process. We will modify the method section to "We used the Marine20 curve for calibration (Heaton et al., 2020), with a time constant ΔR =−95±61 yrs calculated using the Marine20 database (http://calib.org/marine/). This ΔR value was derived from the average reservoir ages of 5 modern bivalve shells from the Laptev Sea (Bauch et al., 2001). The rationale for using an updated ΔR is the ~150 yr shift in the global marine reservoir age between the Marine13 curve used in the previous age model (Hörner et al., 2016) and the Marine20 curve used in this study (Heaton et al., 2023; Heaton et al., 2020)."

Result, Section 4.1 chronology: "The mass accumulation rates (MARs) of all biomarkers were largely affected by the pronounced sedimentation rate changes and thus, showed similar temporal changes in all

terrestrial biomarkers, including HMW n-alkanes, HMW fatty acids, and lignin phenols (Fig S3, contents of each biomarker in Fig S4). Fig 2a and Fig 5 i, j show the mass accumulation rate of HMW fatty acids as a representation.". Mass accumulation rates play a very important role in the environmental interpretations presented in this paper. In a number of areas the author's highlight how important the number of age-depth control points, and more generally changes in sedimentation rate, can impact these calculations. Furthermore, one of the most general questions about how offshore sedimentary processes responded to regional and global climate changes, concerns how the mass accumulation of sediments (sedimentation rate) changed. If there is an influx of material from land, one would expect that there would be enhanced deposition of sediments offshore. One question not answered in the current paper is - do we see this? I think it would be great if the authors added a 'sedimentation rate', or 'mass accumulation rate of sediments' panel in Figures 2 and potentially 5. The age depth models in the supplementary material do not really show the changes in sedimentation rate through time (not in enough detail), and it would be nice to clearly see how this impacting the calculated MAR's of other components (like TerrOM).

We appreciate Reviewer 2's suggestion to incorporate sedimentation rate changes for cores PS51/154 and PS51/159 in Figure S3. This high correlation between elevated terrOM MAR and sedimentation rates is also evident in other pan-Arctic sediment core records (Figure A2-1). We will include this figure in the supplementary as Figure S6.

[Figure]

Fig A2-1. Rate of global sea-level change (Lambeck et al., 2014) and sedimentation rate changes for cores ARA04C/37, Beaufort Sea (Wu et al., 2020); PC23, Laptev Sea (Tesi et al., 2016); PS51/154, Laptev Sea (this study); PS51/159, Laptev Sea (this study); HH11-09, northern Svalbard continental margin (Nogarotto et al., 2023); PS2837-5, Fram Strait (Birgel and Hass, 2004); MSM05/5-712-2, Fram Strait (Müller and Stein, 2014; Aagaard-Sørensen et al., 2014; Zamelczyk et al., 2014). Black triangles indicate the age control points for each record. The blue bars highlight periods of rapid sea-level rise. Meltwater pulses are denoted as mwp-1A and mwp-1B. Paleoclimate periods are abbreviated as HS1: Heinrich Stadial 1, B/A: Bølling-Allerød, YD: Younger Dryas, and PB: Preboreal.

Line 253: "remained rather constant despite the periods of peak MAR (Fig S5)." I think you should specify what MARs you are discussing.

We thank Reviewer 2 for pointing out the ambiguity in the original text. We will revise the sentence as follows: "The values of $\delta^{13}C$ and TOC in cores PS51/154 (−25.6‰ and 0.79%) and PS51/159 (−25.8‰ and 0.99%) remained rather constant between 16.2 and 9.5 kyr BP, despite the three periods of peak MAR of terrestrial biomarkers (Fig S5).".

Line 375: "Before the Bering Strait opened at around 11 kyr BP (Jakobsson et al., 2017), the coastlines of the Beaufort Sea and the Chukchi Sea were connected, allowing the potential westward transport of terrOM from North America. Therefore, we consider the record before 11 kyr BP from the Chukchi Sea (4-PC1) as a representation of terrOM signal from the North American Arctic.". I am not sure that I buy this argument. On one hand this paper is trying to disentangle regional from global climate drivers for TerrOM delivery to the Arctic, but then wants to use a record from the Herald Canyon in the Chukchi Sea as a proxy for deglacial processes operating across Arctic North America? Even when we look at the basic sedimentology, the deglacial records that have been published from the Canadian Beaufort Sea have a very high detrital carbonate content, which is not mirrored in the Chukchi Sea records. Maybe there is something I have misinterpreted, in which case it would be good to clarify what is meant in this sentence.

We agree that linking the Chukchi Sea record to the Canadian Beaufort Sea record is debatable, also the core 4-PC1 record only covers the period between mwp-1A and mwp-1B and the mid-Holocene, rather than the full last deglaciation period. To maintain focus, we will remove the Chukchi Sea record from the discussion.

Line 385: "Age-depth models for these records were recalibrated against the Marine20 calibration curve (Heaton et al., 2020) or a combination of Intcal20 (Reimer et al., 2020) and Marine20 curves, depending on the original studies to achieve congruent age control across all records. Reservoir ages were taken from the original publications." Is this accurate? or were reservoir corrections taken from the original publications and updated to fit with the Marine20 calibration curve by . . . and then specify how this was done.

We thank Reviewer 2 for pointing out the need to update ΔR for the Marine20 calibration curve. This adjustment significantly improves the comparability of terrOM MAR peaks in marine sediment cores to periods of rapid sea-level rise. In Table A2-1, we have compiled the R or ΔR values previously used for each record, and the updated ΔR values for calibration against the Marine20 curve. Following Heaton et al. (2020), we use the Marine20 database to derive updated ΔR values except for core ARA04C/37. For this core, as there are no available ΔR values for the Beaufort Sea or nearby in Marine20 database, we subtract 150 yrs from the ΔR values from the previous age model (Keigwin et al., 2018), as suggested by Heaton et al. (2023). For studies using constant ΔR, we keep the same principle and use the constant updated ΔR for calibration. Three studies are using varying ΔR in their previously published age models: cores ARA04C/37, PC23, and HH11-09. For core ARA04C/37, the new ΔR values for each dating point were calculated by subtracting 150 yrs from the previous ΔR for each dating point (Keigwin et al., 2018). And we referenced the updated Marine20-calibrated age models from Sabino et al. (2024) for core PC23, and from Nogarotto et al. (2023) for core HH11-09, respectively. Table A2-1 will be included in the supplement as Table S3, and we will expand the discussion paragraph about the use of new ΔR. Figure 5 in the manuscript will also be updated accordingly.

Table A2-1. Comparison of previous calibration and updating calibration methods on the core used in this study.

| Core ID | Reference of previous age model | Previous calibration curve | Previous R or ΔR | Updated ΔR for Marine 20 | Method to updated ΔR |
|---|---|---|---|---|---|
| ARA04C/37 JPC15 | Keigwin et al. (2018); Wu et al. (2022) | Marine13 | ΔR=200±100 during younger dryas, 0±100 for the other periods | Variable ΔR, ΔR=50±100 during younger dryas, ΔR=-150±100 for the other periods | Update ΔR from Keigwin et al. (2018) by minus 150 year (Heaton et al., 2023) |
| PC23 | Tesi et al. (2016) | Marine13 for marine samples /Intcal13 for plant samples | ΔR=400 during early Holocene, 67 during mid and late Holocene | ΔR=411±56 during early Holocene, ΔR=−95±91 during mid and late Holocene | Adopted from Sabino et al. (2024) |
| PS51/154 | Taldenkova et al. (2010) | Fairbanks 0107 | R=370, constant | ΔR=−95±61 | From Marine20 database, average of 5 adjacent available datapoints |
| PS51/159 | Taldenkova et al. (2010) | Fairbanks 0107 | R=370, constant | ΔR=−95±61 | From Marine20 database, average of 5 adjacent available datapoints |
| HH11-09 | Nogarotto et al. (2023) | Marine20 | Variable ΔR between each datapoints | Variable ΔR | Adopted from Nogarotto et al. (2023) |
| PS2837-5 | Nørgaard-Pedersen et al. (2003) | CALIB 4.1.2 | R=0±400, constant | ΔR=−41±30 | From Marine20 database, average of 10 adjacent available datapoints |
| MSM5/5-712-2 | Müller and Stein (2014); Aagaard-Sørensen et al. (2014); Zamelczyk et al. (2014) | Marine09 | ΔR=151±51, constant | ΔR=−65±33 | From Marine20 database, average of 7 adjacent available datapoints (distance <620 km) |
| PS2458 | Nicolas et al. (2024) | Marine20 | ΔR=345±60, constant | ΔR=345±60, constant | Adopted from Nicolas et al. (2024) |

Line 407-409:"The rapid global sea-level rise during meltwater pulse 1A (mwp-1A) was an important process in terrOM mobilization across the pan-Arctic region. TerrOM MAR peaks during this period are observed widely in records from the Eurasian Arctic and the Bering Sea". The title of this section is 'Pan-Arctic factor: sea-level rise' but no mention is made here of the Canadian Arctic/Beaufort Sea. I think it is hard to argue this without data from the Canadian Arctic, and it does not seem like that data exists (i.e in Fig 5, the Beaufort Sea records do not extend that far back in time). I can imagine that there may be a difference in glaciated versus non-glaciated margins etc. I at least think that this needs to be discussed in the text, as I am not at all convinced that the Chukchi Sea record is representative of North America. A core from the Herald canyon cannot tell us about all the processes operating across the northern coast of Canada.

We appreciate Reviewer 2's comment. Indeed, the Canadian Arctic suffers from the lack of available records extending before mwp-1A. We agree that the Bering Sea record may not reflect the environmental conditions in the Arctic Ocean, as terrOM sources in the Bering Sea likely originate from Beringia rather than North America. We will exclude the Bering Sea record from the discussion to maintain focus on the Arctic Ocean and modify the section title to "Regionally recurrent factor: sea-level rise" to better align with the scope of the available data.

Lines 431-435: "In the North American Arctic, terrOM MAR peaks appeared during the interval between mwp-1A and mwp-1B (Fig 5d, e). Inland warming in North American began at approximately 13.5 kyr BP, while the Eurasian Arctic remained cold (Brosius et al., 2021). This regional temperature discrepancy possibly explains the exclusive terrOM MAR peaks observed in the North American Arctic during the interval between mwp-1A and mwp-1B (Fig 5c, g)". I think one of the most important observations that is not picked up in this paper is the link between TerrOM fluxes in the Beaufort and Chukchi seas and the d18O excursion reported by Spielhagen et al., 2005 in the outer Laptev Sea. All of these events appear to occur between MWP-1A and MWP-1B and there seems to be a coincidence in timing, even with some of the HMV Mar's in the Laptev Sea (PS51/154) and Fram Strait. However, this is hardly discussed in the paper and I wonder if it deserves more attention (see next comment)

As this comment is directly related to the next comment, we reply to the two comments together below.

Lines 465-470: "However, freshwater events were less likely to be the cause for terrOM MAR in the western Laptev Sea. While freshwater flooding events were recorded in an icedammed lake upstream of the Lena River (14.9 ± 2.0 kyr BP) and in a sediment record from the Laptev Sea (PS2458, at 12.7 ky BP) (Spielhagen et al., 2005; Margold et al., 2018), the timing of these events did not 470 correspond with any of the terrOM MAR peaks in cores PS51/154 and PS51/159. This temporal mismatch suggests that Siberian freshwater pulses had little impact on the increase in terrestrial biomarker MAR in the western Laptev Sea.". The d18O peak in foraminfera from PS2458 (Fig 7 in Spielhagen et al 2005) does seem to overlap or is very close to some of the increased terrigenous biomarkers MAR's shown in figure 5 of this paper. I think it would add a lot to recalibrate the ages of PS2458, and plot this data on one of the summary figures. It seems to be an extremely complementary dataset to the goals of this study (looking at processes impacting terrigenous OC mobilization to the Arctic). The current arguments that it is not correlated to any of the documented periods of enhanced TerrOM flux is not really supported by the current presentation of data. It may be true – but can it be shown more clearly?

We thank Reviewer 2 for the suggestion to recalibrate records from core JPC15 (Beaufort Sea) and core PS2458 (Laptev Sea). The freshwater event identified in the Beaufort Sea (planktic foraminifera δ18O drop in core JPC15)(Keigwin et al., 2018) aligns with terrOM MAR in core ARA04C/37 (Wu et al., 2022; Wu et al., 2020) (Figure A2-2, upper panel). However, in the Laptev Sea, the timing of the freshwater event in core PS2458 did not match the MAR peak in core PS51/154 (Figure A2-2, lower panel). This mismatch suggests that the freshwater event in the Laptev Sea likely did not trigger elevated MAR peaks. We will include a discussion in the revised manuscript and include Figure A2-2 to the supplement as Figure S7.

[Figure]

Figure A2-2. Comparison of freshwater event and terrestrial organic matter (terrOM) mass accumulation rate (MAR) in the Beaufort Sea: (a) δ18O values of *Neogloboquadrina pachyderma* from core JPC15 (Keigwin et al., 2018). (b) Campestreol + β-sitosterol MAR from core ARA04C/37 (Wu et al., 2020). Laptev Sea: (c) δ18O values of *Neogloboquadrina pachyderma* from core PS2458 (Spielhagen et al., 2005)(d) high molecular weight (HMW) fatty acid MAR from core PS51/154 (this study). (e) HMW fatty acid MAR from core PS51/159 (this study). Black triangles denote age control points. Black intervals under MAR peaks indicate age uncertainty ranges. Purple and blue bars highlight freshwater events in the Beaufort Sea and the Laptev Sea, respectively.

**References**

Aagaard-Sørensen, S., Husum, K., Werner, K., Spielhagen, R. F., Hald, M., and Marchitto, T. M.: A Late Glacial–Early Holocene multiproxy record from the eastern Fram Strait, Polar North Atlantic, Marine Geology, 355, 15-26, 10.1016/j.margeo.2014.05.009, 2014.

Bauch, H. A., Mueller-Lupp, T., Taldenkova, E., Spielhagen, R. F., Kassens, H., Grootes, P. M., Thiede, J., Heinemeier, J., and Petryashov, V.: Chronology of the Holocene transgression at the North Siberian margin, Global Planetary Change, 31, 125-139, 2001.

Birgel, D. and Hass, H.: Oceanic and atmospheric variations during the last deglaciation in the Fram Strait (Arctic Ocean): a coupled high-resolution organic-geochemical and sedimentological study, Quaternary Science Reviews, 23, 29-47, 10.1016/j.quascirev.2003.10.001, 2004.

Heaton, T. J., Bard, E., Bronk Ramsey, C., Butzin, M., Hatté, C., Hughen, K. A., Köhler, P., and Reimer, P. J.: A Response to Community Questions on the Marine20 Radiocarbon Age Calibration Curve: Marine Reservoir Ages and the Calibration of 14c Samples from the Oceans, Radiocarbon, 65, 247-273, 10.1017/rdc.2022.66, 2023.

Heaton, T. J., Köhler, P., Butzin, M., Bard, E., Reimer, R. W., Austin, W. E. N., Bronk Ramsey, C., Grootes, P. M., Hughen, K. A., Kromer, B., Reimer, P. J., Adkins, J., Burke, A., Cook, M. S., Olsen, J., and Skinner, L. C.: Marine20—The Marine Radiocarbon Age Calibration Curve (0–55,000 cal BP), Radiocarbon, 62, 779-820, 10.1017/rdc.2020.68, 2020.

Hörner, T., Stein, R., Fahl, K., and Birgel, D.: Post-glacial variability of sea ice cover, river run-off and biological production in the western Laptev Sea (Arctic Ocean)–A high-resolution biomarker study, Quaternary Science Reviews, 143, 133-149, 2016.

Keigwin, L. D., Klotsko, S., Zhao, N., Reilly, B., Giosan, L., and Driscoll, N. W.: Deglacial floods in the Beaufort Sea preceded Younger Dryas cooling, Nature Geoscience, 11, 599-604, 10.1038/s41561-018-0169-6, 2018.

Lambeck, K., Rouby, H., Purcell, A., Sun, Y., and Sambridge, M.: Sea level and global ice volumes from the Last Glacial Maximum to the Holocene, Proceedings of the National Academy of Sciences of the United States of America, 111, 15296-15303, 10.1073/pnas.1411762111, 2014.

Müller, J. and Stein, R.: High-resolution record of late glacial and deglacial sea ice changes in Fram Strait corroborates ice–ocean interactions during abrupt climate shifts, Earth and Planetary Science Letters, 403, 446-455, 10.1016/j.epsl.2014.07.016, 2014.

Nicolas, A., Mollenhauer, G., Lachner, J., Stübner, K., Malter, M., Wollenburg, J., Grotheer, H., and Adolphi, F.: Precise dating of deglacial Laptev Sea sediments via 14C and authigenic 10Be/9Be – assessing local 14C reservoir ages, EGUsphere, 2024, 1-19, 10.5194/egusphere-2024-1992, 2024.

Nogarotto, A., Noormets, R., Chauhan, T., Mollenhauer, G., Hefter, J., Grotheer, H., Belt, S., Colleoni, F., Muschitiello, F., Capotondi, L., Pellegrini, C., and Tesi, T.: Coastal permafrost was massively eroded during the Bølling-Allerød warm period, Communications Earth & Environment, 4, 350, https://doi.org/10.1038/s43247-023-01013-y, 2023.

Nørgaard-Pedersen, N., Spielhagen, R. F., Erlenkeuser, H., Grootes, P. M., Heinemeier, J., and Knies, J.: Arctic Ocean during the Last Glacial Maximum: Atlantic and polar domains of surface water mass distribution and ice cover, Paleoceanography, 18, n/a-n/a, 10.1029/2002pa000781, 2003.

Sabino, M., Gustafsson, Ö., Wild, B., Semiletov, I. P., Dudarev, O. V., Ingrosso, G., and Tesi, T.: Feedbacks From Young Permafrost Carbon Remobilization to the Deglacial Methane Rise, Global Biogeochemical Cycles, 38, 10.1029/2024gb008164, 2024.

Spielhagen, R., Erlenkeuser, H., and Siegert, C.: History of freshwater runoff across the Laptev Sea (Arctic) during the last deglaciation, Global and Planetary Change, 48, 187-207, 10.1016/j.gloplacha.2004.12.013, 2005.

Taldenkova, E., Bauch, H. A., Gottschalk, J., Nikolaev, S., Rostovtseva, Y., Pogodina, I., Ovsepyan, Y., and Kandiano, E.: History of ice-rafting and water mass evolution at the northern Siberian continental margin (Laptev Sea) during Late Glacial and Holocene times, Quaternary Science Reviews, 29, 3919-3935, 2010.

Tesi, T., Muschitiello, F., Smittenberg, R. H., Jakobsson, M., Vonk, J. E., Hill, P., Andersson, A., Kirchner, N., Noormets, R., Dudarev, O., Semiletov, I., and Gustafsson, O.: Massive remobilization of permafrost carbon during post-glacial warming, Nature Communications, 7, 13653, 10.1038/ncomms13653, 2016.

Wu, J., Stein, R., Fahl, K., Syring, N., Nam, S.-I., Hefter, J., Mollenhauer, G., and Geibert, W.: Deglacial to Holocene variability in surface water characteristics and major floods in the Beaufort Sea, Communications Earth & Environment, 1, 10.1038/s43247-020-00028-z, 2020.

Wu, J., Mollenhauer, G., Stein, R., Kohler, P., Hefter, J., Fahl, K., Grotheer, H., Wei, B., and Nam, S. I.: Deglacial release of petrogenic and permafrost carbon from the Canadian Arctic impacting the carbon cycle, Nature Communications, 13, 7172, 10.1038/s41467-022-34725-4, 2022.

Zamelczyk, K., Rasmussen, T. L., Husum, K., Godtliebsen, F., and Hald, M.: Surface water conditions and calcium carbonate preservation in the Fram Strait during marine isotope stage 2, 28.8–15.4 kyr, Paleoceanography, 29, 1-12, 2014.

---

## Author Response (AR1)

**Revision overview on cp-2024-60**

We would like to thank the editor and the two reviewers for their precious suggestions, which significantly helped us to improve this manuscript. Following a suggestion from one of our co-authors, we replaced the Sd/Sl ratio with the Vd/Vl ratio to provide a better overview of vegetation degradation. This change was made because syringyl phenols (Sd and Sl) are predominantly found in angiosperm vascular plants, while vanillyl phenols (Vd and Vl) are present in both gymnosperm and angiosperm vascular plants (cf. Tesi et al., 2014). The correlation between the Sd/Sl and Vd/Vl ratios is high in both cores (R = 0.97, p < 0.0001 in core PS51/154, and R = 0.94, p < 0.0001 in core PS51/159, see the supplementary figure below this revision overview), so this substitution did not affect the interpretation of the data.

In the revised version, we include an additional degradation index using *n*-alkanes to supplement the Vd/Vl ratio, adopted the $C_{25}/(C_{25}+C_{29})$ ratio as a peat input index, and updated ΔR values to calibrate all the core records to the Marine20 curve. These changes in indices and ΔR values did not significantly alter the results, and the interpretation remain generally unchanged. We also excluded records from non-Arctic regions and those with lower temporal resolution to maintain focus on Arctic records. The format was checked, including reordering the in-text citations by year and double-checking the overall formatting.

Below, we outline manuscript revisions corresponding to each comment. Comments from the editor and the reviewers are in blue, and our responses are in black.

Dear Tsai-Wen Lin and co-authors,

Than you for submitting your responses to the reviewer comments. I invite you to resubmit your manuscript with all responses incorporated. Please submit both a track changes and a clean version of the updated manuscript, together with an overview of how the responses have been incorporated. I keep the option open to obtain a new evaluation after revision.

From your response to Reviewer 1, comment 2, it is not clear for me if you plan to include clarification/discussion on this point in the revised version, following the argumentation in your response. I will suggest that you do.

We thank the editor's suggestion to include this part of the discussion. Please refer to our response to Reviewer 1, Comment 2.

I have one other minor comment that you may want to consider. In the introduction you say that "Studying these periods of rapid environmental change can improve the understanding of how current abrupt warming, sea ice loss, and sea-level rise might affect permafrost stability and the release of previously freeze-locked carbon". I cannot see that you follow up in this statement in your discussion/conclusion. Is it possible to add a sentence addressing implications of your results for our understanding of current changes?

We added a paragraph to the conclusion addressing projected permafrost mobilization under warming scenarios in lines 510–517.

Best regards,

Bjørg Risebnrobakken
Editor, Climate of the Past

**Revisions according to comments by reviewer 1 on cp-2024-60**

In this manuscript, Lin et al., evaluate changes in terrestrial organic matter input into the Arctic ocean over the last 18,000 years. The chosen study sites drain vast quantities of permafrost, such that enhanced terrestrial OC input might indicate enhanced permafrost thaw. The authors identify several pulses of terrestrial OC burial, some of which may correlate to known climatic events (e.g, meltwater pulse 1a). There is remarkably low terrOC input during the Holocene but lots of variability in terrOC delivery between 10 to 16 kyr. Each peak shows different compositional characteristics, suggesting distinct terrOM sources derived from different mechanisms. However, it would be interesting to try and unravel this further using your existing data (see comments below). Overall, the manuscript is well written, the figures are clear and the captions offer an appropriate level of detail. A few comments are included below that would be worth exploring further...

Comment 1: it is hard to tell whether the pulse in terrestrial OC is due to permafrost thaw or enhanced delivery of plant/soil OC. This could be assessed by measuring 14C values in different lipids, but I realize is beyond the scope of this paper. However, there is existing literature that could be helpful - for example, Feng et al. (2013; PNAS) explored how the 14C signature of different compound classes varied across the Pan-Arctic region and lignin phenols appear to mostly derive from recent carbon, whereas n-alkanes are derived from older carbon sources (e.g,. permafrost). Although you don't have 14C measurements, it could imply distinct sources for lignin vs n-alkanes in your samples.
...but another way to tackle this could be to look at other n-alkane indices such as the carbon preference index. This is frequently used to assess changes in OC maturity and may provide further insights into the type of terrOC that is being delivered into the marine realm, especially during the three pulses. If it was older OC, it may yield slightly lower CPI values. It may also tell you whether you are reworking old petrogenic OC into the marine realm too (which could be a CO2 source if it was oxidised; see work by Sparkes 2016 the Cryosphere, but also work by Bob Hilton/Valier Galy etc)

The calculation of the CPI index has been added in lines 212−217, and the CPI data is included in Fig 2c. The CPI results for cores PS51/154 and PS51/159, along with their source implication for permafrost rather than petrogenic sources, are discussed in lines 264−271. The discussion of how transport distance affects CPI data is included in lines 352−363. As noted in our response to the reviewer's comment, the ratio of lignin phenol to long-chain lipid contents in core PS51/154 was more influenced by a slumping event than by paleoenvironmental changes, so we didn't include the index.

Comment 2: Export vs preservation
It is important to confirm that the increase in leaf wax mass accumulation rates is not due to enhanced preservation but is reflecting enhanced terrOC export. This could be explored by calculating MARs of of short-chain alkanes (algal-derived), mid-chain alkanes (moss or macrophyte derived), and long-chain alkanes (vascular plant) n-alkanes during the three "pulses". If all three increase, it might suggest that OC preservation is the main driver. But if only the mid- and long-chain alkane MARs increase, it would imply enhanced terrOC input.

The discussion about terrOC export and preservation has been included in lines 320−330. Content changes for short-chain, mid-chain, and long-chain (HWM) *n*-alkanes are presented in Fig S6.

Comment 3: a note on pAq

The authors use the pAq index (the ratio between mid vs long chain n-alkanes) to infer changes in wetland input, but I would note that's its more complex than this. For example, both Sphagnum moss and aquatic macrophytes are characterized by similar lipid distributions (Baas et al., 2000; Ficken et al., 2000), so without knowledge of the local vegetation, its challenging to say whether the pAq ratio is due to changes in moss input or macrophyte input. Perhaps you could narrow this down by drawing upon predicted vegetation patterns during the Holocene/LGM etc. There is also great work by Jorien Vonk, Bart van Dongen, Orjan Gustafsson etc in similar pan-arctic regions that might be useful. For example, Vonk 2009 paper in Org. Geochem shows that "…the C25/(C25 + C29) n-alkane ratio is most suitable for terrestrial OM source apportionment in these coastal regions". This might be worth exploring alongside the pAq (although I suspect you will get similar results!).

References to vegetation changes and peatland development during the Preboreal and early Holocene in northern Siberia have been added in lines 342−344. We adopted the $C_{25}/(C_{25}+C_{29})$ *n*-alkane ratio, with the calculation method in lines 203−211 of the revised manuscript. Results are described in lines 262−265, and Fig 2b has been updated accordingly. The interpretation remains the same as that of the previously used $P_{aq}$ index due to the identical results between the two indices.

Minor comments:
L63: v/v – the v's should be italicised – change throughout

This has been corrected in lines 163, 166, and 170 in the revised manuscript.

L205: correct that its used for macrophytes – but also for sphagnum mosses.

The introduction of the $P_{aq}$ index has been replaced with an introduction to the $C_{25}/(C_{25}+C_{29})$ *n*-alkane ratio, as in lines 203−210 of the revised manuscript.

L252: is this statistically significant?

The modified sentence is in lines 257−258 of the revised manuscript. While the content remains unchanged, the sentence clarity has been improved.

L365: dialkyl

Please refer to the next response.

L366: and also found in peats – and lakes - and marine sediments!
The original sentence introducing brGDGTs has been removed from the revised manuscript, as the terrOM MAR record using brGDGT is not from the Arctic Ocean and has been excluded from the revised discussion.

L412: subscript CO2

This is corrected in line 441 of the revised manuscript.

**Revisions according to comments by reviewer 2 on cp-2024-60**

Summary

This paper presents organic geochemical proxies used to distinguish the source of organic matter (OM) in two 17-18 kyr records from the Laptev sea. The data is integrated with pan-Arctic records in an attempt to investigate local versus global drivers of enhanced OM flux during deglaciation. It is a very nice study and paper that fits well within the scope of Climate of the Past. However, I also think there is some room for improvement in he presentation and discussion of the data. Specifically, 1) a clearer presentation of sedimentation rate changes in the studied cores (and how they impact the mass accumulation rate estimates) is important, 2) a more in-depth discussion about the timing and link to OM flux/source changes associated with previously identified meltwater events in the Laptev Sea between MWP-1A and MWP1B. These are discussed further in the points below:

Specific comments
Line 25: "Additional terrOM MAR peaks coincided with periods of enhanced inland warming, prolonged ice-free conditions, and freshwater flooding, which varied between regions.". I wonder if this sentence could be more specific, for instance, be specific about what periods evidence for coastal erosion in response to sea-level rise are identified, and at what periods are inland warming and freshwater flooding seen? Also - maybe the last sentence could be re-written so that the expression 'regional terrOM fluxes' only occurs once.

The revised abstract, as proposed in the comment response, is presented in lines 22–26 of the revised manuscript.

Could the authors add a bit more information on how a dR value of -95 +/- 65 years for the Marine20 calibration curve was derived from Bauch et al, 2001? It is good to describe the conversion of old dR values when making them compatible with the Marine20 calibration curve, it is good bookkeeping.

Detailed information about the updated ΔR values for cores PS51/154 and PS51/159 has been incorporated in lines 140–145 of the revised manuscript.

Result, Section 4.1 chronology: "The mass accumulation rates (MARs) of all biomarkers were largely affected by the pronounced sedimentation rate changes and thus, showed similar temporal changes in all terrestrial biomarkers, including HMW n-alkanes, HMW fatty acids, and lignin phenols (Fig S3, contents of each biomarker in Fig S4). Fig 2a and Fig 5 i, j show the mass accumulation rate of HMW fatty acids as a representation.". Mass accumulation rates play a very important role in the environmental interpretations presented in this paper. In a number of areas the author's highlight how important the number of age-depth control points, and more generally changes in sedimentation rate, can impact these calculations. Furthermore, one of the most general questions about how offshore sedimentary processes responded to regional and global climate changes, concerns how the mass accumulation of sediments (sedimentation rate) changed. If there is an influx of material from land, one would expect that there would be enhanced deposition of sediments offshore. One question not answered in the current paper is - do we see this? I

think it would be great if the authors added a 'sedimentation rate', or 'mass accumulation rate of sediments' panel in Figures 2 and potentially 5. The age depth models in the supplementary material do not really show the changes in sedimentation rate through time (not in enough detail), and it would be nice to clearly see how this impacting the calculated MAR's of other components (like TerrOM).

Sedimentation rates of cores PS51/154 and PS51/159 have been added to Fig S3. Additionally, changes in sedimentation rate across Arctic records are now included in Fig S7.

Line 253: "remained rather constant despite the periods of peak MAR (Fig S5)." I think you should specify what MARs you are discussing.

The modified sentence is in lines 257–258 of the revised manuscript.

Line 375: "Before the Bering Strait opened at around 11 kyr BP (Jakobsson et al., 2017), the coastlines of the Beaufort Sea and the Chukchi Sea were connected, allowing the potential westward transport of terrOM from North America. Therefore, we consider the record before 11 kyr BP from the Chukchi Sea (4-PC1) as a representation of terrOM signal from the North American Arctic.". I am not sure that I buy this argument. On one hand this paper is trying to disentangle regional from global climate drivers for TerrOM delivery to the Arctic, but then wants to use a record from the Herald Canyon in the Chukchi Sea as a proxy for deglacial processes operating across Arctic North America? Even when we look at the basic sedimentology, the deglacial records that have been published from the Canadian Beaufort Sea have a very high detrital carbonate content, which is not mirrored in the Chukchi Sea records. Maybe there is something I have misinterpreted, in which case it would be good to clarify what is meant in this sentence.

Record from the Chukchi Sea has been excluded from the revised discussion and Fig 5 to maintain better focus on comparing terrOM MAR peaks with periods of rapid sea-level rise.

Line 385: "Age-depth models for these records were recalibrated against the Marine20 calibration curve (Heaton et al., 2020) or a combination of Intcal20 (Reimer et al., 2020) and Marine20 curves, depending on the original studies to achieve congruent age control across all records. Reservoir ages were taken from the original publications." Is this accurate? or were reservoir corrections taken from the original publications and updated to fit with the Marine20 calibration curve by . . . and then specify how this was done.

The selection of updated ΔR values for calibrating against the Mairne20 curve in each core is described in lines 409–417 and Table S3 of the revised manuscript. Age models in Fig 5 have been updated accordingly. After updating to the new age models, the terrOM MAR peaks align more closely between cores, as well as between core records and the two periods of rapid sea-level rise. The discussion for core ARA04C/37 from the Canadian Arctic has been modified as described in lines 441–445, while interpretations of other records remain unchanged.

Line 407-409:"The rapid global sea-level rise during meltwater pulse 1A (mwp-1A) was an important process in terrOM mobilization across the pan-Arctic region. TerrOM MAR peaks during this period are observed widely in records from the Eurasian Arctic and the Bering Sea". The title of this section is 'Pan-

Arctic factor: sea-level rise' but no mention is made here of the Canadian Arctic/Beaufort Sea. I think it is hard to argue this without data from the Canadian Arctic, and it does not seem like that data exists (i.e in Fig 5, the Beaufort Sea records do not extend that far back in time). I can imagine that there may be a difference in glaciated versus non-glaciated margins etc. I at least think that this needs to be discussed in the text, as I am not at all convinced that the Chukchi Sea record is representative of North America. A core from the Herald canyon cannot tell us about all the processes operating across the northern coast of Canada.

Records from the Bering Sea have been excluded from the revised discussion to focus solely on the Arctic Ocean. The section title of Chapter 5.2.1 in line 435 has been modified to "Regionally recurrent factor: sea-level rise".

Lines 431-435: "In the North American Arctic, terrOM MAR peaks appeared during the interval between mwp-1A and mwp-1B (Fig 5d, e). Inland warming in North American began at approximately 13.5 kyr BP, while the Eurasian Arctic remained cold (Brosius et al., 2021). This regional temperature discrepancy possibly explains the exclusive terrOM MAR peaks observed in the North American Arctic during the interval between mwp-1A and mwp-1B (Fig 5c, g)". I think one of the most important observations that is not picked up in this paper is the link between TerrOM fluxes in the Beaufort and Chukchi seas and the d18O excursion reported by Spielhagen et al., 2005 in the outer Laptev Sea. All of these events appear to occur between MWP-1A and MWP-1B and there seems to be a coincidence in timing, even with some of the HMV Mar's in the Laptev Sea (PS51/154) and Fram Strait. However, this is hardly discussed in the paper and I wonder if it deserves more attention (see next comment)

As this comment is directly related to the next one, we have combined our responses below.

Lines 465-470: "However, freshwater events were less likely to be the cause for terrOM MAR in the western Laptev Sea. While freshwater flooding events were recorded in an icedammed lake upstream of the Lena River (14.9 ± 2.0 kyr BP) and in a sediment record from the Laptev Sea (PS2458, at 12.7 ky BP) (Spielhagen et al., 2005; Margold et al., 2018), the timing of these events did not 470 correspond with any of the terrOM MAR peaks in cores PS51/154 and PS51/159. This temporal mismatch suggests that Siberian freshwater pulses had little impact on the increase in terrestrial biomarker MAR in the western Laptev Sea.". The d18O peak in foraminfera from PS2458 (Fig 7 in Spielhagen et al 2005) does seem to overlap or is very close to some of the increased terrigenous biomarkers MAR's shown in figure 5 of this paper. I think it would add a lot to recalibrate the ages of PS2458, and plot this data on one of the summary figures. It seems to be an extremely complementary dataset to the goals of this study (looking at processes impacting terrigenous OC mobilization to the Arctic). The current arguments that it is not correlated to any of the documented periods of enhanced TerrOM flux is not really supported by the current presentation of data. It may be true – but can it be shown more clearly?

The comparison between freshwater events and terrOM MAR peaks in the Beaufort Sea and the Laptev Sea is presented in lines 482–492 and Fig S8 of the revised manuscript. All the records used for this comparison have been calibrated against the Marine20 curve, and the updated ΔR values are listed in Table S3.

**Supplementary figure**

[Figure]

Figure 1. Sd/Sl (black lines) and Vd/Vl (blue lines) ratios for cores (**a**) PS51/154 and (**b**) PS51/159, and correlations between Sd/Sl and Vd/Vl ratios in cores (**c**) PS51/154 and (**d**) PS51/159.

**Reference**

Tesi, T., Semiletov, I., Hugelius, G., Dudarev, O., Kuhry, P., and Gustafsson, Ö.: Composition and fate of terrigenous organic matter along the Arctic land–ocean continuum in East Siberia: Insights from biomarkers and carbon isotopes, Geochimica et Cosmochimica Acta, 133, 235-256, 10.1016/j.gca.2014.02.045, 2014.